# Genetic parallels in biomineralization of the calcareous sponge *Sycon ciliatum* and stony corals

**Oliver Voigt[1]\*, Magdalena V Wilde[1,2], Thomas Fröhlich[2], Benedetta Fradusco[1], Sergio Vargas[1], Gert Wörheide[1,3]**

[1]Department of Earth and Environmental Sciences, Paleontology and Geobiology, Ludwig Maximilians-Universität München, Munich, Germany; [2]Gene Center—Laboratory for Functional Genome Analysis, Ludwig-Maximilians-Universität München, Munich, Germany; [3]GeoBio-Center, Ludwig-Maximilians-Universität München, Munich, Germany

## eLife Assessment

This **important** paper reports the discovery of calcarins, a protein family that seems to be involved in calcification in the calcareous sponge *Sycon ciliatum*, significantly enhancing our understanding of the molecular and cellular mechanisms underlying spicule formation in sponges and the evolution of carbonate biomineralization. The conclusions are supported by **compelling** evidence based on an integrated analysis that combines transcriptomics, genomics, proteomics, and precise *in situ* hybridization. These findings will be of broad interest to cell biologists, biochemists, and evolutionary biologists.

**\*For correspondence:**
oliver.voigt@lmu.de

**Competing interest:** The authors declare that no competing interests exist.

**Abstract** The rapid emergence of mineralized structures in diverse animal groups during the late Ediacaran and early Cambrian periods likely resulted from modifications of pre-adapted biomineralization genes inherited from a common ancestor. As the oldest extant phylum with mineralized structures, sponges are key to understanding animal biomineralization. Yet, the biomineralization process in sponges, particularly in forming spicules, is not well understood. To address this, we conducted transcriptomic, genomic, and proteomic analyses on the calcareous sponge *Sycon ciliatum*, supplemented by *in situ* hybridization. We identified 829 genes overexpressed in regions of increased calcite spicule formation, including 17 calcarins—proteins analogous to corals' galaxins localized in the spicule matrix and expressed in sclerocytes. Their expression varied temporally and spatially, specific to certain spicule types, indicating that fine-tuned gene regulation is crucial for biomineralization control. Similar subtle expression changes are also relevant in stony coral biomineralization. Tandem gene arrangements and expression changes suggest that gene duplication and neofunctionalization have significantly shaped *S. ciliatum*'s biomineralization, similar to that in corals. These findings suggest a parallel evolution of carbonate biomineralization in the calcitic *S. ciliatum* and aragonitic corals, exemplifying the evolution of mechanisms crucial for animals to act as ecosystem engineers and form reef structures.

## Introduction

The evolution of biomineralization allowed animals to produce important mineralized functional structures, such as shells, teeth, and skeletons. For this, they most frequently use calcium carbonate, calcium phosphate, and silicate as mineral components, of which calcium carbonate is taxonomically

**eLife digest** Many animals – from corals to snails to humans – build hard structures such as shells, teeth, and skeletons through a process called biomineralization. In this process, specialized cells deposit minerals like calcium carbonate with the help of dedicated proteins and their genes.

Biomineralization has evolved independently across different animal lineages, often by reusing or modifying existing genes. Among the earliest animals to produce mineralized structures were sponges, many of which form skeletal elements known as spicules. Calcareous sponges, for example, build spicules from calcite, but the underlying genetic and molecular mechanisms remain poorly understood. By contrast, stony corals, which produce aragonite skeletons, have a relatively well-characterized "biomineralization toolkit." Studying how sponges construct their spicules, therefore, offers a unique opportunity to explore the origins and diversity of biomineralization strategies across animals.

To address this, Voigt et al. investigated the genes and proteins involved in forming calcitic spicules in the calcareous sponge Sycon ciliatum. This species is an excellent model because each spicule is produced by only a few specialized cells, enabling precise analysis of gene expression during biomineralization in a simple system.

Using a combination of transcriptomic, genomic, and proteomic approaches, and a method called RNA *in situ* hybridization, the team identified 829 genes that were upregulated in spicule-forming areas in the sponge. Among them were 17 calcarins – a newly described group of galaxin-like proteins (which are proteins also found in corals) that were expressed in mineral-secreting cells known as sclerocytes and incorporated into the spicule matrix.

These proteins appear to be unique to calcareous sponges and absent from other sponge lineages. Different spicule types showed distinct patterns of gene activation, suggesting specialized roles. Evidence of gene duplication and neofunctionalization – that is, duplicated genes adopting a new role – was also found. Together, these results suggest molecular parallels between the calcitic spicules of sponges and the aragonitic skeletons of corals, despite their independent evolutionary origins.

These findings deepen our understanding of the evolution and regulation of biomineralization – a process central to marine ecosystems. They may inform future research into how reef-building organisms such as corals and sponges respond to environmental challenges like climate change and ocean acidification. Insights into how organisms control mineral formation could inspire biomimetic materials and sustainable synthesis techniques. Future studies will be needed to explore the functional roles of these genes and their ecological significance.

the most diverse (*Knoll, 2003*; *Murdock and Donoghue, 2011*). Animal biominerals are usually composite materials of organic and mineral compounds and exhibit material properties and shapes that differ enormously from their purely mineral counterparts. Their formation requires biological control governed by specific genes and proteins in specialized biomineralization cells and tissues. Those then ensure supply with the necessary inorganic substances into the calcifying space (intracellularly or extracellularly) to enable crystal precipitation and growth, e.g., by pH regulation. They also provide organic components, including secreted, so-called 'matrix proteins' that influence the polymorph and shape of the biomineral and subsequently may become embedded in it (*Aizenberg et al., 1996*; *Aizenberg et al., 1994*).

The ability to form biominerals evolved several times independently in animal lineages, and early instances of mineralized animal skeletons appeared within a geologically brief span, from the latest Ediacaran to the Middle Cambrian (*Murdock and Donoghue, 2011*). According to the 'biomineralization toolkit' hypothesis, animals of this era had acquired pre-adapted genes and gene-regulatory networks necessary to produce biominerals in a controlled manner that were further refined independently in different lineages (*Murdock, 2020*). This can be achieved by gene family expansion, in which, after initial gene duplication, diversification by mutation and natural selection leads to the neofunctionalization of a copied gene in the biomineralization process.

Among early branching, non-bilaterian calcifiers, stony corals are best studied, and essential elements of their genetic biomineralization machinery are known that originate from gene duplication and neofunctionalization, such as the SLC4γ bicarbonate transporter (SLC4γ) that is a key component

of the coral aragonite skeleton forming machinery (*Tinoco et al., 2023*; *Zoccola et al., 2015*). Numerous skeletal matrix proteins have been identified (*Drake et al., 2013*; *Peled et al., 2020*; *Ramos-Silva et al., 2013*). Among those, galaxin and related proteins were the first to be characterized, and at least in some species, comprise the most dominant protein in skeletal matrix extractions (*Fukuda et al., 2003*; *Watanabe et al., 2003*). Acidic proteins are another essential component of coral skeletal matrices, presumably influencing the polymorph of the precipitating carbonate (*Drake et al., 2013*; *Laipnik et al., 2020*; *Mass et al., 2016*). However, the genetic mechanisms of the calcium carbonate biomineralization in sponges (class Porifera) are less known. Sponges produce skeletal elements in the form of differently shaped spicules, siliceous in the extant sponge classes Demospongiae, Hexactinellida, and Homoscleromorpha, and calcitic only in the class Calcarea (calcareous sponges). Nonetheless, some demosponges like the polyphyletic 'sclerosponges' may form a rigid calcium carbonate basal skeleton in addition to or instead of siliceous spicules (*Wörheide, 1998*). Essential skeletal matrix proteins occluded in such rigid calcitic carbonate skeletons of sclerosponges have been studied in *Astrosclera willeyana* (*Jackson et al., 2011*; *Jackson et al., 2007*) and *Vaceletia* sp. (*Germer et al., 2015*). In contrast to those aragonite-producing sclerosponges, calcareous sponges continuously produce calcitic spicules of different shapes, providing a unique opportunity to observe all stages of the process and study differences in their production. Additionally, their spicule formation is often faster than the production of skeletal elements in other calcifiers. It takes only a few days from initiation to the finished spicule and involves only a few specialized cells called sclerocytes (*Ilan et al., 1996*; *Woodland, 1905*). They control biomineralization by their relative movement to each other and by secreting specific proteins, ions, and other substances into the calcification space, a process still little understood at the molecular level. Only a few genes involved in calcification in

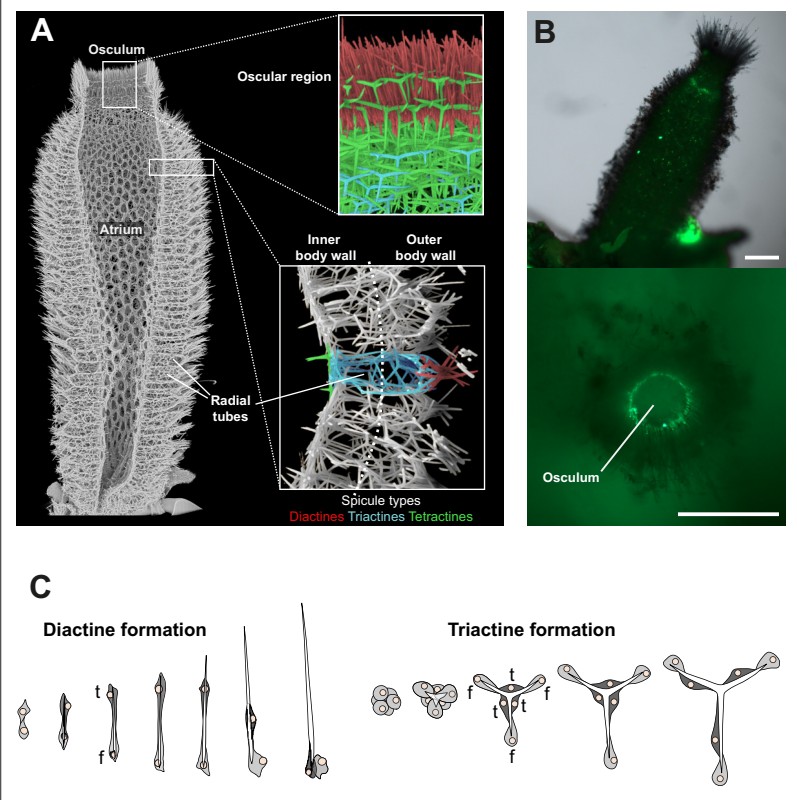

**Figure 1.** Skeletal organization and spicule formation in *S. ciliatum*. (**A**) The *S. ciliatum* skeleton features specific spicule types in distinct body regions: parallel diactines in the oscular region (upper inset), radial tubes supported by triactines and tufted with diactines (lower inset), and the atrial skeleton composed of triactines and tetractines. (**B**) The upper oscular region shows increased spicule formation (calcein staining) in the growing zone of new radial tubes and around the osculum, where oscular diactines are predominantly produced (modified from *Voigt et al., 2014*). Scale bars: 0.5 mm. (**C**) Spicules are formed by sclerocytes, specialized cells controlling spicule formation. Diactine formation involves two sclerocytes, triactine formation six (f=founder cell, t=thickener cell).

calcareous sponges are known, including two specific carbonic anhydrases, and two SLC4 bicarbonate transporters, and three acidic proteins, two of which are spicule-type specific (*Voigt et al., 2021*; *Voigt et al., 2017*; *Voigt et al., 2014*). Skeletal matrix proteins have not yet been identified but are of particular interest because they may play a crucial role in spicule morphogenesis (*Aizenberg et al., 1995*).

In this study, we used genomic data, differential gene expression (DGE) analysis, and RNA *in situ* hybridization (ISH) experiments, supplemented by a proteomic approach, to identify genes directly involved in the spicule formation in the calcarean model species *Sycon ciliatum* (subclass Calcaronea), and compare those with another non-bilaterian calcifying clade, the scleractinian (stony) corals, the ecosystem engineers of today's coral reefs. We found surprising similarities in key components of the biomineralization toolkit between calcareous sponges and corals that shed new light on the evolution of calcium carbonate biomineralization in animals.

## Results

*Sycon ciliatum* is a tube-shaped sponge with a single apical osculum and a sponge wall of radial tubes around the central atrium (*Figure 1A*). The radial tubes are internally lined with choanoderm, which forms elongated chambers in an angle of approximately 90° to the tube axis. The sponge tissue is supported by a skeleton composed of countless spicules arranged in a specific pattern within the sponge body (*Figure 1A*). The spicules are formed by sclerocytes located within the mesohyl, the extracellular matrix of the sponge. Connected by septate junctions, they enclose an extracellular space where a spicule grows (*Ledger, 1975*). Only a small number of sclerocytes participate in forming a single spicule (*Figure 1C*). Production of each actine (ray) of a spicule involves a pair of sclerocytes (*Woodland, 1905*): a founder cell, which initiates actine elongation at the tip, and a thickener cell, located alongside the actine. The thickener cell enhances actine strength in some species by precipitating additional calcium carbonate. More spicules are produced in the apical growth zone than in other regions of *Sycon's* body (*Voigt et al., 2014*). In this zone, diactine spicules, one-rayed spicules with two pointed tips, are arranged in a palisade-like manner and continuously grow around the osculum opening. Simultaneously, new triactines, three-rayed spicules resembling a Mercedes star, and four-rayed tetractines form in the atrial skeleton. Large amounts of triactines and distal diactine bundles are also produced within newly developing radial tubes (*Figure 1B*).

### DGE analysis

We performed a DGE analysis comparing gene expression of the apical oscular region to the inner and outer body walls of the more basal body regions in five specimens (*Appendix 1—table 1*). We observed 1575 differentially expressed genes (log2-fold change ≥2, padj<0.01). Of these genes, 829 were overexpressed in the oscular region, including the known biomineralization genes with documented sclerocyte-specific expression: CA1, CA2, AE-like1, AE-like2, Diactinin, Triactinin, Spiculin (*Voigt et al., 2017*; *Voigt et al., 2014*).

To gain further insight into these genes' potential function, we performed a GO-term enrichment analysis, considering the GO-term annotation of the closest hit in Uniprot against each transcript as a potential annotation for the *S. ciliatum* genes. The genes overexpressed in the oscular regions were enriched in biological process GO-terms relevant for biomineralization (e.g. with the representative terms GO:0001501 'skeletal system development', GO:0001503: 'ossification', GO:0030282 'bone mineralization', *Supplementary file 1*). Among them were genes similar to Fibrilin-1, Collagen alpha-1 (XXVIII) chain B, Collagen alpha-1 (I) chain, Collagen alpha-1 (XI) chain, V-type proton ATPase subunit a, and to Fibroblast growth factor receptor 2. Additionally, the *S. ciliatum*-specific growth factor SciTgfBH, with similarity to bone morphogenetic protein 2, and the homeobox protein SciMsx, with similarity to the homeobox protein MSX-2, belong to genes with these enriched GO terms. Other enriched GO terms related to ongoing morphogenesis, cell differentiation, cell signaling, cell migration, and organization of the organic matrix are all indicative of the growth zone that the oscular region represents (*Supplementary file 1*). Noteworthy among them are genes of the Wnt pathway (SciWntG, I, J, K, L, M, O, U, SciFzdD, and Metalloprotease TIKI homolog).

Fourteen secreted proteins, with significantly higher expression in the oscular region, showed similarity to the coral skeletal organic matrix proteins galaxin and galaxin-like proteins, first described from

the coral *Galaxea fascicularis* (*Fukuda et al., 2003*; *Watanabe et al., 2003*). In total, 17 proteins similar to galaxins were identified from the predicted proteins of the *S. ciliatum* genome using BLASTp. We refer to these as calcarin 1–17 (Cal1–Cal17) to discriminate them from galaxin and galaxin-like proteins of scleractinian corals. The presence of signal peptides indicates that calcarins are secreted. Except for a Reeler domain (Pfam PF02014) in Cal16, calcarins lack recognizable domains. Like galaxins, calcarins contain regions with 10–23 di-cysteine residues, typically separated by 10–15 amino acids. Additionally, a single cysteine occurs approximately at the inter-di-cysteine distance upstream and downstream of the di-cysteine region. Monomer AlphaFold predictions propose that disulfide bridges between these cysteines provide a tertiary structure typical of calcarins and galaxins (*Figure 2*; *Figure 2— figure supplement 1*). The di-cysteines form the N-terminal part of a common four-amino acid beta-hairpin with a short, often two-amino acid long turn. The first cysteine of a di-cysteine connects with the second cysteine of the preceding beta-hairpin by a disulfide bridge, linking the beta-hairpins together. This disulfide bridge backbone is continuous in galaxins, whereas the predictions in calcarins may show one or two interruptions.

## Temporal and spatial expression of calcarins

The expression of Cal1 to Cal8 was investigated using chromogenic *in situ* hybridization (CISH) and hairpin chain reaction fluorescence *in situ* hybridization (HCR-FISH), confirming their presence in sclerocytes (*Figure 3*). Additionally, antisense probes were used against the mRNA of Spiculin, an acidic protein specific to all thickener cells, Triactinin, a marker for triactine and tetractine thickener cells, and the sclerocyte-specific carbonic anhydrase SciCA1 (*Voigt et al., 2017*; *Voigt et al., 2014*). This approach enabled us to visualize and contextualize calcarin-positive cells (*Figure 3—figure supplement 1*). The expression patterns of calcarins varied with sclerocyte type and spicule formation stage.

Cal1 is predominantly observed in the founder cells of triactines and tetractines and only minimal overlap with Cal2 expression (*Figure 3A–D*). Initially expressed in all six founder cells, expression ceases in the central sclerocytes after they transform into thickener cells. The Cal1 signal persists in the remaining founder cells at the tips of the growing spicules (*Figure 3D and E*). In the founder cells of the diactines, Cal1 is only expressed transiently, as is evident by the lack of Cal1 signal in most oscular diactine founder cells (*Figure 3A and B*). Occasionally, we observed co-expression of Spiculin and Cal1 in diactine sclerocytes, presumably during the onset of the conversion of a diactine founder cell to a thickener cell (*Figure 3E*). At slightly later stages (recognizable by the elongated cell shape), Cal1 is no longer expressed in the thickener cells. At even later stages, Cal1 expression also ceases in the founder cells of the diactines, which instead express Cal2 (*Figure 3E*, *Figure 3—figure supplement 1*). The expression of Cal2 and Spiculin can be recognized in two superimposed rings of cells around the osculum, representing the founder cells and the associated thickener cells of the oscular diactines (*Figure 3B*).

Cal3, Cal7, and Cal8 are produced by all founder cells, regardless of the type of spicule they form (*Figure 3F–I and L*). During the transformation into thickener cells, the expression of these calcarins ceases and is replaced by the expression of Spiculin. Again, co-expression of some of these calcarins with Spiculin was observed in a few cells, probably in sclerocytes transitioning from founder to thickener cells (*Figure 3G*). Cal4 is expressed in thickener cells, Cal5 only in thickener cells of triactines, like Spiculin and Triactinin, respectively (*Figure 3J*). Cal6 expression mirrors that of Cal2, occurring in rounded cells at the distal tip of radial tubes and in a ring of cells around the oscular ring (*Figure 3K*). In these regions, Spiculin-expressing cells are spatially associated. In regions without diactines, like the atrial cavity, no Cal6 expression occurs, even if Spiculin expression indicates ongoing production of triactines and tetractines (*Figure 3—figure supplement 2*). Like Cal2, Cal6 must be expressed by later-stage diactine founder cells, which appear more spherical than elongated. At the end of radial tubes, Cal2 and Cal6 positive founder cells are in contact with choanocytes (*Figure 3—figure supplement 2A*). In summary, calcarin expression in sclerocytes varies across spicule formation stages and between sclerocytes of different spicule types (*Figure 4*), adding further complexity to the spatiotemporal expression of biomineralization genes that was reported before (*Voigt et al., 2017*). Notably, in young coral polyps, normalized and scaled expression data of the raw cell counts of 980 calicoblastic cells indicate nonoverlapping expression of two galaxin-like proteins, an uncharacterized skeletal matrix protein and a skeletal-aspartic acid-rich protein (*Appendix 2—figure 1*), suggesting spatiotemporal expression changes of skeletal matrix proteins also occur in calicoblastic cells of corals.

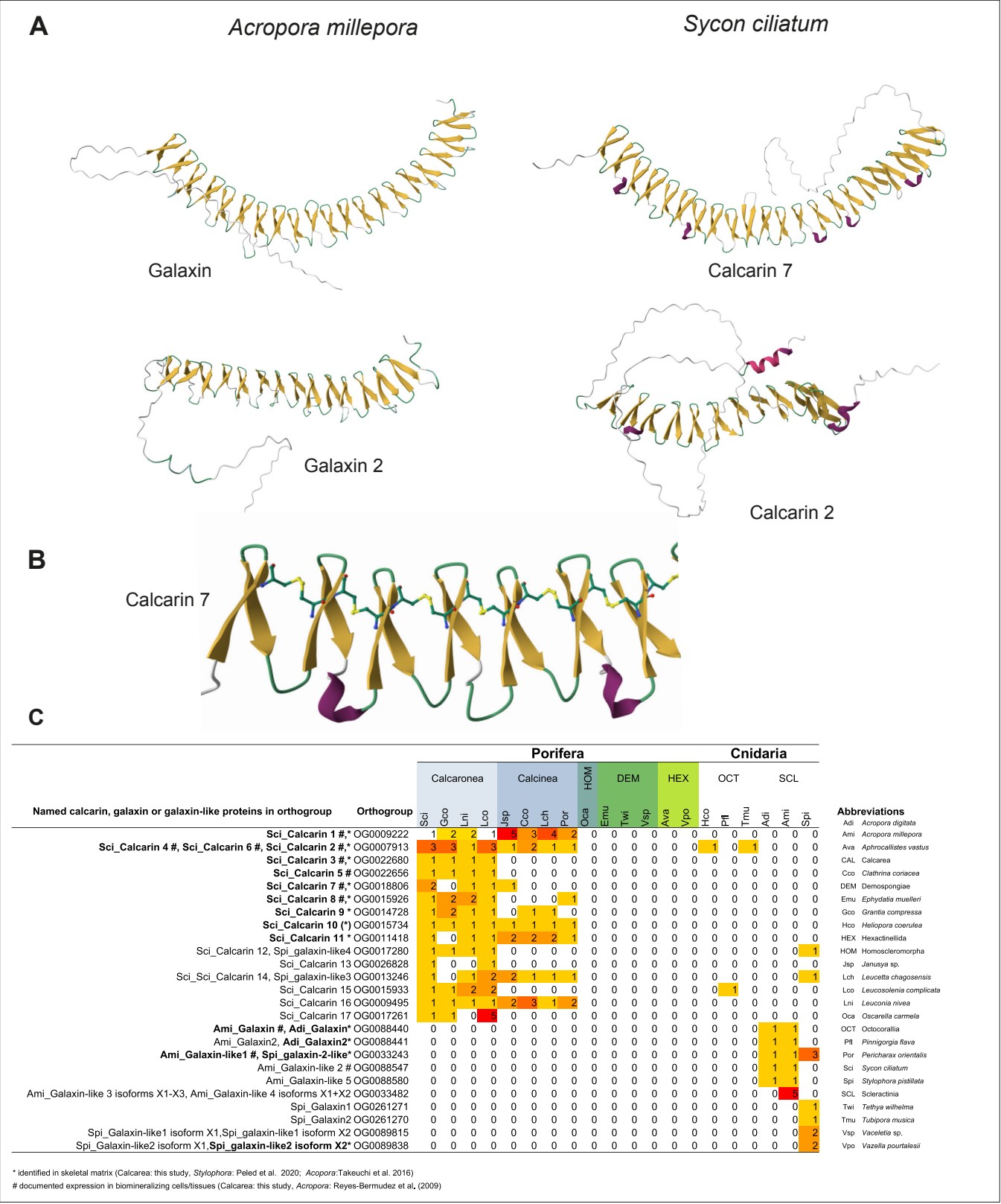

**Figure 2.** Calcarin and galaxin predicted structures and occurrences in sponges and corals. (**A**) Structural similarities in AlphaFold predictions of galaxins (*A. millepora*) and selected calcarins (*S. ciliatum*). (**B**) Beta-hairpins in Cal7 connected by disulfide bridges of di-cysteines. (**C**) Number of calcarins, galaxin-like, and galaxins transcripts in sponges and corals, assigned to orthogroups. Additional AlphaFold structure predictions of selected *S. ciliatum* calcarins and coral galaxin-like proteins are provided in *Figure 2—figure supplements 1 and 2*.

*Figure 2 continued on next page*

*Figure 2 continued*

The online version of this article includes the following figure supplement(s) for figure 2:

**Figure supplement 1.** AlphaFold structure predictions of additional selected *S. ciliatum* calcarins.

**Figure supplement 2.** AlphaFold predictions of galaxin-like proteins from octocorals and stony corals that fall in orthogroups with *S. ciliatum* calcarins.

## Calcarins and other proteins are embedded in the spicule matrix

We extracted spicules from the tissue using sodium hypochlorite to examine the proteins incorporated in the biomineral. The spicules were then washed with water and dissolved in acetic acid. Mass spectrometry was employed to analyze the acid-insoluble proteins, while the acid-soluble ones were too diluted for adequate examination. With the obtained spectra, we identified 35 proteins (1.0% FDR protein threshold, with at least two peptides per protein, *Supplementary file 2*). Fifteen of these proteins are encoded by overexpressed transcripts in the oscular region (*Figure 5A*; *Supplementary file 2*). We consider these proteins to be specialized biomineralization proteins. Several calcarins expressed in founder cells (Cal1, Cal2, Cal3, Cal7, Cal8) were most prominent. Additional calcarins of the skeletal matrix were Cal9, Cal11, and Cal10, although the latter was only supported by one peptide.

In addition, we found three secreted proteins without any domain or other prominent features other than a signal peptide, which we refer to as skeletal matrix proteins 1–3 (SMP 1–3). Other spicule matrix proteins contain recognizable domains. Two proteins, similar to cysteine-rich secretory protein LCCL domain-containing 2-like (CAH0893205) and peptidase inhibitor 15-like protein (CAH0891878), both belong to the CAP superfamily. Further, we detected a fibronectin III-domain-containing protein (CAH0869470), a subtilisin-like protease (CAH0797261), and an EGF-like domain-containing protein (CAH0821874). Both the LCCL domain-containing 2-like protein and the fibronectin III-domain-containing protein are acidic with isoelectric points of 4.34 and 4.75, respectively. Thickener cell-specific proteins, such as Cal4, Cal5, or the Asx-rich proteins Triactinin and Spiculin, were not observed in the spicule matrix.

When comparing the 15 spicule matrix proteins of *Sycon* to the skeletal matrix proteins of a *Vaceletia* sp. (*Germer et al., 2015*), a demosponge with a calcium carbonate (aragonite) basal skeleton, their similarity is limited to subtilisin-like proteases found in both proteomes. Additional similarity can be seen only in eight additional proteins of presumably intracellular origin (annotated as or containing domains of Histones, Actin, Tubulin, Elongation factor 1a, mitochondrial ATP synthase alpha, *Supplementary file 3*), which were not overexpressed in the oscular region of *S. ciliatum*. These proteins may represent ubiquitous extracellular matrix components or, particularly in the case of proteins with intracellular origins, could result from contamination by residual cellular material or their incorporation into the forming biomineral, as suggested before (*Ramos-Silva et al., 2013*; *Germer et al., 2015*), and we do not consider them to have specific function in biomineralization.

## Expression of biomineralization genes during whole-body regeneration

Dissociated cells of *S. ciliatum* form spherical cell aggregations that differentiate into small functional sponges within 16–18 days. This regeneration resembles the post-larval development in *S. ciliatum* after the first days (*Soubigou et al., 2020*). The first stages are free of spicules, and the first diactines appear 3–4 days after reaggregation. Spicule production, including triactines and tetractines, increases in the following days. After about 16 days, an osculum is formed and framed by diactines. Reanalyzing raw RNA-seq data covering the complete regeneration process (*Appendix 1—table 2*; *Soubigou et al., 2020*), we find downregulated expression of calcarins 1–13, the other spicule-matrix proteins, and previously identified sclerocyte-specific genes (*Voigt et al., 2017*; *Voigt et al., 2014*) in regeneration stages with no or on-setting spicule formation (days 1–4) compared to later stages (days 6–18) with higher numbers of newly forming spicules (*Figure 5B*). Noteworthy, dissociated cells (day 0) also have low expression of these sclerocyte-specific genes, suggesting that most sclerocytes do not survive the cell-dissociation process.

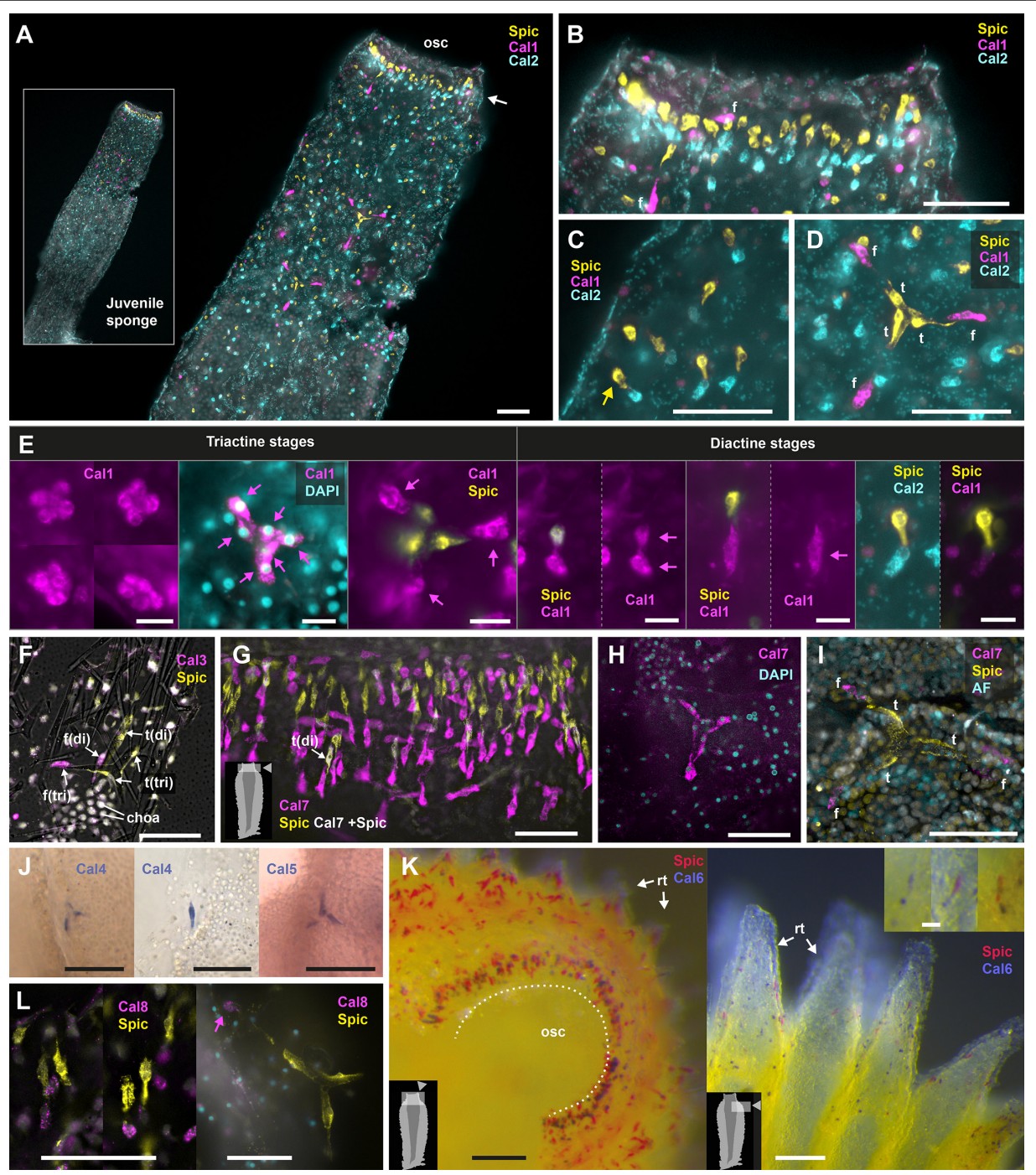

**Figure 3.** Expression of calcarins. Insets indicate the location of the depicted view within the sponge body, where applicable. To improve accessibility for individuals with red/green color vision deficiency, original RGB channel colors (*Figure 3—figure supplement 3*) were modified to a cyan/magenta/blue color scheme. AF: autofluorescence; osc = osculum; rt: radial tubes; Spic: Spiculin (**A–D**) Cal1, Cal2, and Spiculin expression in a regenerated *S. ciliatum* at the asconoid juvenile stage. Scale bars = 50 μm. (**A**) Overview of the entire specimen, highlighting distinct gene expression with minimal co-expression in the apical half of the sponge. Arrow points to the ring of founder and thickener cells that form the oscular diactines. (**B**) Detailed view on the expression around the oscular region; Spiculin in thickener cells (apical), Cal2 in diactine founder cells (basal), and Cal1 in triactine/tetractine founder cells (f). (**C**) Sponge wall detail; Cal2 in diactine founder cells, Spiculin in thickener cells (arrow: one diactine thickener cell). (**D**) Triactine/tetractine founder cells expressing Cal1, thickener cells expressing Spiculin. (**E**) Cal1 expression in founder cells ceases as they transform into thickener cells, and Spiculin expression sets in. Cal1 continues to be expressed in actine-producing founder cells in triactines, but in the diactine actine-forming founder cell, it is replaced by Cal 2 expression in later stages. Scale bars; 10 μm. (**F**) Expression of Cal3 in founder cells and Spiculin in thickener cells attached to the preserved diactine (di) and triactine (tri) spicules (overlay with light microscopic image). Note how thickener cells thinly ensheath the spicule.

*Figure 3 continued on next page*

*Figure 3 continued*

Scale bar: 50 µm. (**G**) Cal7 expression in the founder cells of oscular diactines. Co-expression of Spiculin and Cal7 rarely occurs in transient stages of emerging thickener cells. Scale bar: 50 µm. (**H**) Early triactine stage with six founder cells expressing Cal7. Scale bar: 50 µm. (**I**) In later triactine formation stages, thickener cells no longer express Cal7. Scale bar: 50 µm. (**J**) Expression of Cal4 and Cal5 in thickener cells. Scale bar: 50 µm. (**K**) Cal6 and Spiculin expression in oscular region (osc: oscular opening) and expression at the distal end of radial tubes (rt) of the body wall. Scale bars = 100 µm, inset 20 µm. (**L**) Expression of Cal8 in founder cells and Spiculin in thickener cells of diactines at the end of radial tubes (left) and of a triactine (right).

The online version of this article includes the following figure supplement(s) for figure 3:

**Figure supplement 1.** Expression of biomineralization genes in radial tubes (rt).

**Figure supplement 2.** Expression of Cal2 and Cal6.

**Figure supplement 3.** Version of *Figure 3* with the original RGB channels of the fluorescent images (**A–I** and **L**).

## The majority of calcareous sponge biomineralization genes show concerted changes in expression in different biological settings

We performed a weighted gene co-expression network analysis (WGCNA) of the body part and regeneration datasets to identify co-expression modules representing groups of genes displaying concerted expression patterns. The analysis provided eight meta-modules, of which four showed significant changes in expression module eigengenes—summary profiles that capture the overall expression pattern of each module—between samples with high spicule formation context (osculum region and regeneration stages older than 4 days) and samples with low spicule formation (sponge wall and early regeneration stages until days 3–4) (*Appendix 2—figure 2*). One meta-module 'midnight-blue' showed higher module eigengene expressions in the context of higher spicule formation. The module includes 196 genes, all differentially expressed in the body part dataset. Of these, 189 genes

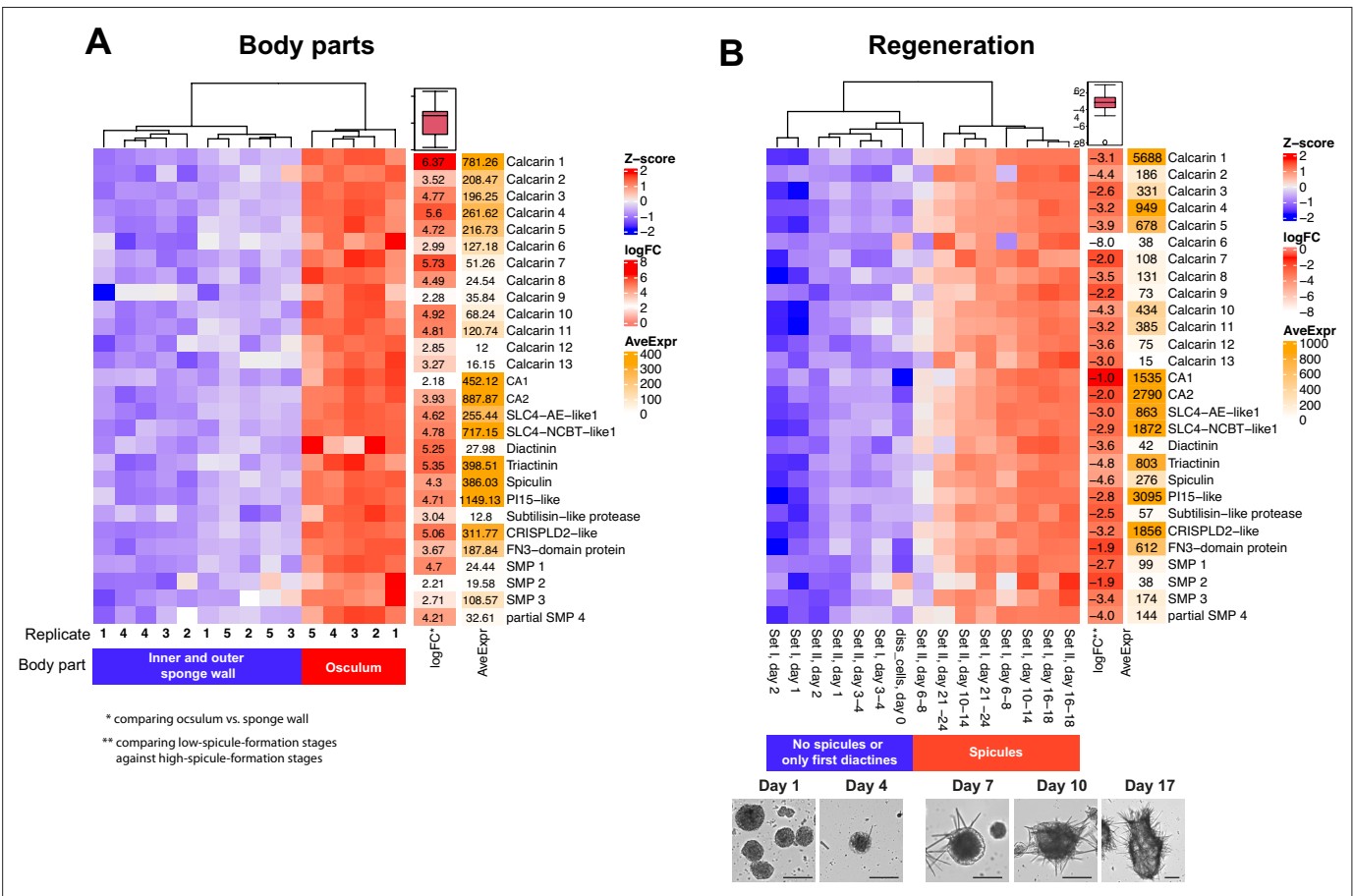

**Figure 5.** Differential gene expression of 13 calcarins and other confirmed or candidate biomineralization genes. (**A**) Osculum region vs sponge wall. (**B**) Changes in relative expression during whole-body regeneration. Scale bars: 100 µm.

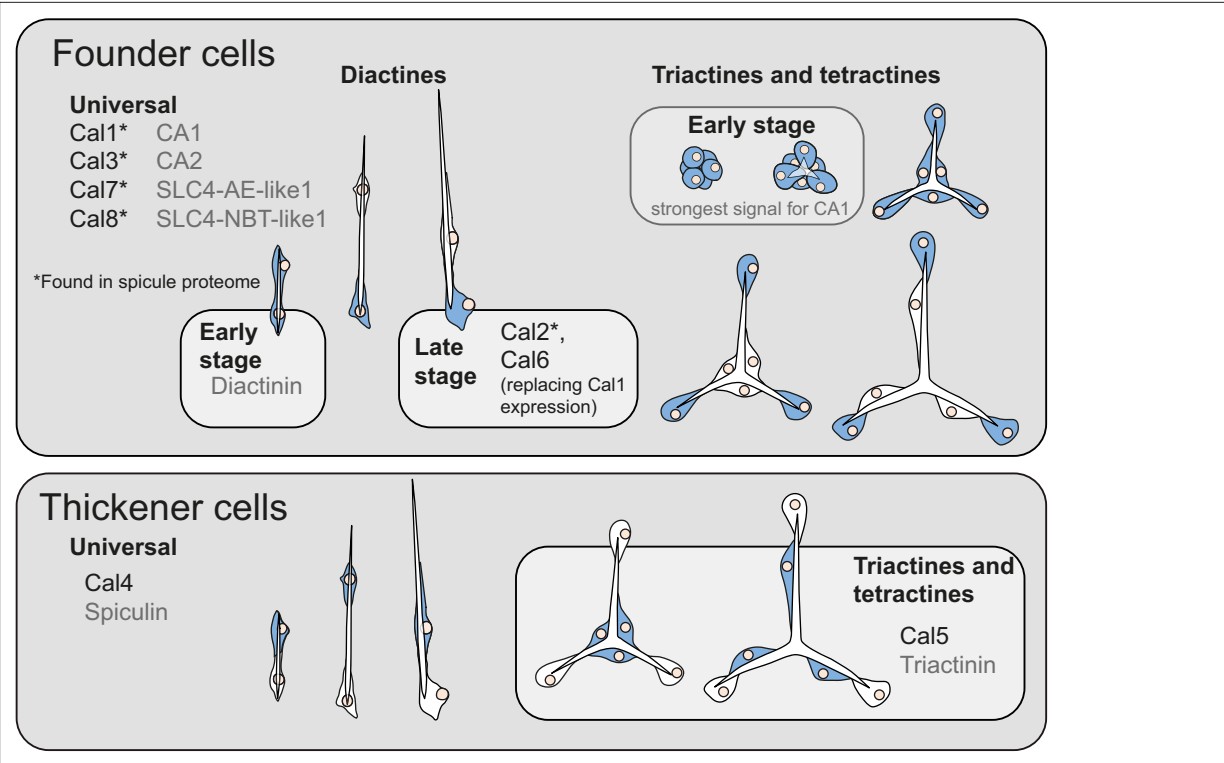

**Figure 4.** Summary of expression changes of biomineralization genes in sclerocytes (expressing cells in blue). In initial spicule formation stages, all sclerocytes act as founder cells. Genes with expression patterns described previously (*Voigt et al., 2017*; *Voigt et al., 2014*) are shown in gray.

were overexpressed in the oscular region, including the suggested biomineralization genes, except for Cal6, Cal9, Cal13, and SMP2. Only seven genes in this meta-module are underexpressed in the oscular region. Some enriched biological process GO terms of genes in this meta-module could be relevant in the context of spicule formation, e.g., monoatomic anion transport, cell junction organization and assembly, and cell-to-cell signaling (*Appendix 2—figure 3*). Eight proteins in the meta module 'midnightblue' are involved in the regulation of transcription (GO:0006355), including SciMsx and other transcription factors. The meta module also contains five genes annotated as similar to Integrin alpha 8 or Paxillin from the transforming growth factor beta signaling pathway (GO:0007179), and three genes annotated to belong to the Wnt signaling pathway (GO:0016055), namely SciFzdD, SciDvlB, and SciWntI (*Supplementary file 4*).

## Calcarins and related galaxin-like proteins in other species

To identify potential homologs of calcarins in a range of other species, we employed OrthoFinder to analyze transcriptomes from calcareous sponges, other sponge classes, octocorals, and stony corals (*Appendix 1—table 3*). This analysis segregated 14 *S. ciliatum* calcarins into distinct orthogroups while grouping Cal2, Cal4, and Cal6 in a single orthogroup. Calcarins in each orthogroup were found in the transcriptomes of other calcareous sponges, either exclusively in Calcaronea (Cal3, Cal5, Cal13, Cal15, and Cal17) or in both Calcaronea and Calcinea (*Supplementary file 5*). Within the same orthogroup, Cal2, Cal4, and Cal6 exhibited sequence identities of 25–36% among themselves, while each showed a higher match in *Grantia compressa*, with identities ranging from 44% to 58%.

Two orthogroups, one containing Cal2, Cal4, Cal6, and the other Cal15, also included galaxin-like sequences from octocorals. Orthogroups containing Cal12 and Cal14 include sequences from the genome of the scleractinian coral *Stylophora pistillata*, but not from *Acropora millepora*. Despite their presence in the same orthogroups, the octocoral and stony coral proteins were only distantly related to the calcareous sponge calcarins (e.g. 12–24% identity between octocoral and calcareous sequences in orthogroup Cal 2-4-6), resulting in poor sequence alignment. AlphaFold predictions suggest that the galaxin-like proteins from the octocorals *Heliopora coerulea* and *Tubipora muscia* share a similar

structure with Cal2, 4, and 6, each featuring 12 beta-hairpins (*Figure 2—figure supplement 2*). In contrast, galaxin-like proteins from *Stylophora* exhibit a substantially higher number of beta-hairpins than the *S. ciliatum* calcarins that are part of the same orthogroup (*Figure 2—figure supplement 2*). Furthermore, none of these scleractinian and octocoral proteins included in the calcarins orthogroup have been directly linked to biomineralization: they were not detected in these species' skeletons (*Conci et al., 2020*, *Peled et al., 2020*) nor have their expression patterns been characterized. Their homology to calcarins, therefore, remains to be determined. Finally, no other sponge proteins grouped with the 17 identified calcarins from *S. ciliatum*, although a few galaxin-like sequences were detectable using BLASTp (*Supplementary file 6*)—precisely one in Homoscleromorpha, one in each Hexactinellida species, and four in the transcriptome, but not the skeletal proteome (*Germer et al., 2015*), of the demosponge *Vaceletia* sp. While the galaxin-like transcripts of *Vaceletia* are short and incomplete, the hexactinellid and homoscleromorph proteins are complete and are large proteins of about 5000 amino acids. A signal peptide suggests that they, too, are secreted. In contrast to calcarins or galaxins, they contain multiple recognizable domains, such as several fibronectin type III and laminin EGF domains. Only about 200 amino acids of these proteins are made from the di-cysteine-containing region of the Galaxin BLASTp hit, which lies between fibronectin type III domains (*Appendix 2—figure 4*). The demosponges *Ephydatia muelleri* and *Tethya wilhelma* lack galaxin-like sequences in their genomes.

Galaxins and galaxin-like proteins are attributed to a PANTHER 'family' PTHR34490, and in InterPro (https://www.ebi.ac.uk/interpro/), 623 proteins are currently annotated to this family (with 381 available AlphaFold structures, see *Supplementary file 6*). They occur in several animal phyla, but also other eukaryotes and archaeans. Species without skeletons, such as the cnidarians *Hydra*, *Actinia*, *Exaiptasia*, and *Nematostella*, also possess galaxin-like proteins assigned to the PANTHER 'family' PTHR34490 (*Supplementary file 6*).

## Sequential order and expression of biomineralization genes

In the *S. ciliatum* genome, the 17 calcarins occur on chromosomes 1, 2, 4, 7, and 13. On chromosome 2, Cal12, Cal4, Cal6, and Cal2 are positioned sequentially (*Figure 6* and *Supplementary file 7*). Cal4, Cal6, and Cal2 belong to a single orthogroup, indicating their homology. Cal4 is expressed in thickener cells, while Cal2 and Cal6 have identical expression patterns, suggesting they are produced by diactine founder cells during the late diactine formation stage (see above). Cal12 has low expression levels, and we did not locate its expression. The founder cell-specific carbonic anhydrase gene CA2 is situated on the reverse strand of chromosome 4, flanked by CA8 and CA3 on the forward strand.

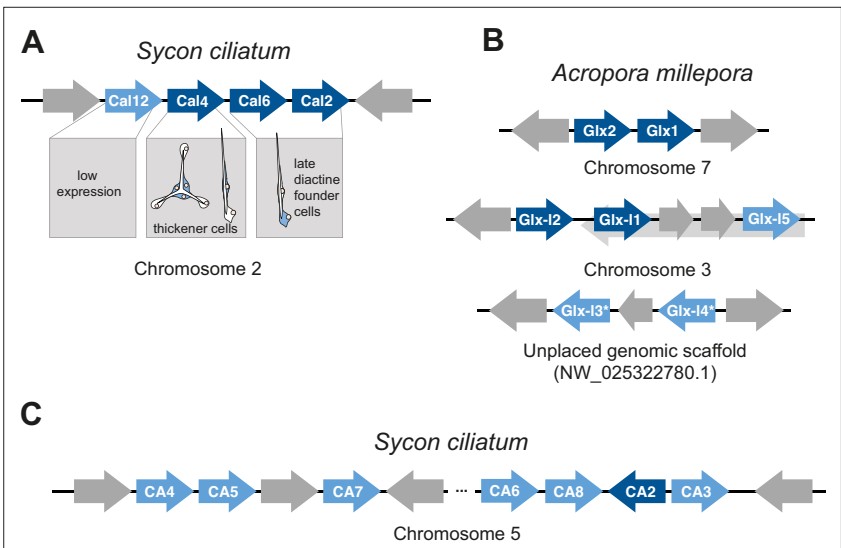

**Figure 6.** Arrangements of biomineralization genes (dark blue) and related genes (lighter blue). *Predicted nested genes not shown. (**A**) Calcarins (Cal) in *S. ciliatum*. (**B**) Galaxin (Glx) and Galaxin-like (Glx-l) proteins in the stony coral *A. millepora*. (**C**) Membrane-bound carbonic anhydrases (CA) in *S. ciliatum*.

Upstream on the same chromosome, other membrane-bound carbonic anhydrases (*Voigt et al., 2021*) are present, including CA4, CA5, CA6, and CA7.

In the coral *A. millepora*, Galaxin1 and Galaxin2 are located subsequently on the genome, making a past gene duplication likely as well. The galaxin-like proteins 1 and 3 also occur in subsequent pairs or triplets. The genes for galaxin-like 2, 1, and 5, as well as galaxin-like 3 and 4, are separated by a maximum of two short predicted genes (*Figure 6*).

## Discussion

### Calcarins are galaxin-like biomineralization proteins

Our study significantly expanded the known non-bilaterian biomineralization genes and found that galaxin-like calcarins are key components of the calcareous sponge biomineralization machinery. Most calcarins could be linked to biomineralization by their higher expression levels in body parts or regeneration stages with increased spicule formation, sclerocyte-specific expression, presence in the extracted spicule matrix, or a combination of these. Additionally, they change their expression in concert with other previously characterized biomineralization genes.

In stony corals and octocorals, certain galaxins and galaxin-like proteins are part of the coral skeletal organic matrix (*Fukuda et al., 2003*; *Watanabe et al., 2003*) or expressed briefly before the onset of biomineralization in the primary polyp (*Reyes-Bermudez et al., 2009*). Typically, one or two galaxins occur in stony coral (*Peled et al., 2020*; *Ramos-Silva et al., 2013*; *Takeuchi et al., 2016*) or octocoral (*Conci et al., 2020*; *Le Roy et al., 2021*) skeletons. In the spicule matrix of *S. ciliatum*, however, we found seven to eight calcarins, so these galaxin-like matrix proteins are much more diverse in calcareous sponges than in corals.

### Role of calcarins, galaxins, and galaxin-like proteins

The temporal and spatial differences in expression patterns of calcarins suggest specialized functions in controlling the biomineralization process. The observed expression patterns of calcarins show that initially, all sclerocytes involved in the formation of a single spicule are functional founder cells expressing the same repertoire of biomineralization genes (e.g. expression of Cal1 and Cal7, *Figure 3E and H*). Founder cell-specific calcarins are integrated into the spicule matrix, much like galaxins and galaxin-like proteins are in coral skeletons. In corals, galaxins and galaxin-like proteins may provide either a structural framework for the growing biomineral or might influence the nucleation and growth of the carbonate crystals (*Fukuda et al., 2003*; *Watanabe et al., 2003*), although their function is not fully understood. We assume similar roles for the calcarins we identified from the spicule matrix. A recent study has shown that calcareous sponge spicules grow by addition of calcium carbonate granules that form near the membranes of active sclerocytes (*Wendt et al., 2025*). Secreted calcarins may play a role in the formation of these granules and are likely among the previously unidentified intercalated proteins that were supposed to influence crystal growth in calcareous sponges by selectively inhibiting growth in specific directions (*Aizenberg et al., 1996*). We hypothesize that differences in calcarin composition in different spicule types influence their specific crystallographic features and possibly their macromorphology (*Aizenberg et al., 1995*; *Aizenberg et al., 1994*).

Our results suggest thickener cells emerge from founder cells by changing gene expression, including calcarins. Thickener cell-specific calcarins and acidic proteins are not found in the spicule matrix, implying that *S. ciliatum* thickener cells do not add significant calcite material to the spicule. Instead, biomineralization genes secreted by thickener cells (Cal4, Cal5, acidic proteins Triactinin and Spiculin) may affect only the spicule's surface. For instance, spicules in one Calcinea species have a calcitic core surrounded by amorphous calcium carbonate cored again by a thin calcitic sheath (*Aizenberg et al., 2003*). Here, acidic proteins and calcarins might be involved in transforming amorphous calcium carbonate to calcite, but due to the thin nature of the sheath, they are more likely to be lost (due to dissolution during clean-up) or remain below the detection threshold.

### The calcareous sponge biomineralization toolkit resembles that of corals

Our widened understanding of the calcareous sponge biomineralization machinery revealed that the identified effector genes are distinct from those reported from other sponges. This is not surprising

because even the formation of siliceous spicules in the classes Demospongiae and Hexactinellida shows little overlap (*Francis et al., 2023*; *Shimizu et al., 2024*), which indicates that the formation of spicules in these three extant sponge groups evolved independently. Based on our data, we can also exclude that there are major genetic similarities between the formation of calcite spicules of calcareous sponges and the aragonitic basal skeletons of some demosponges, called sclerosponges. Only specific subtilisin-like protease and carbonic anhydrases are shared in the biomineralization machinery of the sclerosponge *Vaceletia* and the calcarean *S. ciliatum*. Both belong to larger gene families, and phylogenetic analyses of carbonic anhydrases suggest that they were independently recruited for biomineralization in calcareous sponges and carbonate-producing demosponges (*Voigt et al., 2021*; *Voigt et al., 2014*). Spherulin, a skeletal protein of aragonitic sclerosponges (*Germer et al., 2015*; *Jackson et al., 2011*), is not present in the calcareous sponge genome. Although we identified galaxin-like sequences in *Vaceletia's* transcriptome, the proteins are not reported from its skeleton (*Germer et al., 2015*). In contrast, the galaxin-like calcarins are diverse in calcareous sponges and their spicules.

Considering these genetic differences in sponge biomineralization between taxa that secrete aragonite (*Astrosclera*, *Vaceletia*) and calcite (*Sycon*), it is unexpected that the essential biomineralization effector genes of calcareous sponges that produce calcite resemble those utilized by aragonitic stony corals, suggesting parallel evolution in gene recruitment between these calcifying organisms. The gene families involved are key parts of the pre-adapted carbonate 'biomineralization toolkit' of their common ancestor, including essential pH regulators and for the directional transport of inorganic carbon, i.e., specialized carbonic anhydrases (*Bertucci et al., 2013*; *Voigt et al., 2014*) and bicarbonate transporters (*Voigt et al., 2017*; *Zoccola et al., 2015*; *Appendix 2—figure 5*). Galaxin and galaxin-like proteins from corals and calcarins from calcareous sponges extend the set of similar biomineralization genes.

## Possible origin of galaxins, galaxin-like proteins, and calcarins

The high sequence divergence within calcarins, galaxin, and galaxin-like proteins makes a resolved phylogenetic analysis of all these genes impossible. The likeliest origin of these biomineralization proteins is the recruitment of secreted di-cysteine-rich proteins with other functions and probably different origins. The characteristic feature of these proteins is the more or less evenly distributed di-cysteines, which, according to the AlphaFold prediction, establish a similar tertiary structure with beta-hairpins connected via disulfide bridges. Secreted proteins with this structure are not limited to calcifying organisms and can be portions of larger proteins. For example, in hexactinellid and homoscleromorph sponges, the few proteins similar to calcarins are large and contain numerous functional domains, while the typical di-cysteine region only resembles a small proportion of the protein. Clearly, then, galaxin-like proteins in non-calcifying organisms have other functions.

Although our results clearly show that some calcarins have a common origin, it is unclear whether galaxin-like proteins generally have common or multiple convergent origins. For example, the di-cysteine regions of galaxin and galaxin-like proteins of the Scleractinia consist of more or less conserved repeat motifs of different lengths (*Fukuda et al., 2003*; *Reyes-Bermudez et al., 2009*). These galaxin-like proteins may result from repeated duplication of an original motif, gradually extending the di-cysteine region. In contrast to the galaxin-like proteins of corals, the calcareous sponge calcarins do not show such detailed repeat structures, which could indicate a different origin.

## Gene duplication gave rise to and diversified biomineralization genes in calcareous sponges and corals

Our results provide evidence about the origin and diversification of biomineralization genes. The similarity and sequential arrangement of Cal4, Cal6, and Cal2 on chromosome 2 in the *S. ciliatum* genome suggest that these genes originated from two successive gene duplications. Their different expression patterns reveal that a functional change must have occurred after the gene duplication. Accordingly, the first duplication event produced two copies, leading to one gene specialized for thickener cells and another for late diactine founder cells. Subsequently, the latter gene duplicated again, resulting in the origin of Cal2 and Cal6, which, despite beginning to diverge in sequence, maintain identical expression patterns. This sequence of events highlights the role of gene duplication in the functional diversification of calcarin genes in *S. ciliatum*.

Similar observations for galaxins and galaxin-like genes of the stony coral *A. millepora* suggest that gene duplication is a typical pattern in galaxins and galaxin-like proteins. Galaxin1 and Galaxin2 also occur adjacently in the coral's genome, indicating past gene duplication. The clustering of galaxin-like proteins in pairs or triplets and the separation of genes for Galaxin-like2, 4, and 5, as well as Galaxin-like3 and 4, by a maximum of two short predicted genes, may also be evident for their origin by past duplication events. The fact that one of the *S. ciliatum* sclerocyte-specific carbonic anhydrases, SciCA2, is likewise located on the reverse strand of chromosome 4 and flanked by two non-biomineralizing carbonic anhydrases (SciCA8 and SciCA3) (*Voigt et al., 2014*) encoded on the forward strand provides additional support that gene duplication followed by a neofunctionalization of SciCA2 is the origin of this biomineralization gene. A similar process, tandem gene duplication, neofunctionalization, and inversion, gave rise to the coral-specific biomineralization-specific bicarbonate transporter SLC4γ (*Tinoco et al., 2023*). Our data show that similar genes in both lineages were independently duplicated, and one copy was recruited in the biomineralization machinery by a functional shift and expression in calcifying cells. The example of the three calcarins provides evidence of how the biomineralization process can evolve to become more and more fine-tuned by further gene duplications and subfunctionalization, e.g., in our case, becoming specific for a distinct spicule type.

## Gene regulatory networks

While our study identified biomineralization genes at the effector level, more research is required to understand the underlying gene regulatory networks, e.g., what triggers the expression changes in some of the spicule's sclerocytes to change from functioning as a founder cell to a thickener cell. The genes that change expression in meta-module midnightblue might include some key players, as they are enriched in some regulatory biological process GO terms (*Appendix 2—figure 3*). Cell-cell signaling between the cooperating sclerocytes could play a critical role in controlling the temporal change in gene expression, and several apically overexpressed genes reported here may be involved, such as components of the Wnt or BMP signaling pathways, involved in mammal bone morphogenesis and homeostasis (*Sánchez-Duffhues et al., 2015*; *Zhong et al., 2014*). However, what these are and how the control mechanisms do function must be investigated in further studies because, in both biological scenarios, apical gene expression or regeneration, additional growth and differentiation processes run parallel to spicule formation, which we cannot differentiate from each other with the methods used here.

## Conclusion

We have identified calcarins, galaxin-like proteins, as novel components of the biomineralization toolkit in calcareous sponges. The diversity of calcarins in the calcitic sponge spicules, compared to galaxin and galaxin-like proteins in aragonitic coral skeletons, underscores their specialized roles in biomineralization. Their varied spatial and temporal expression patterns highlight the precise genetic regulation involved in calcification in this class of sponges. Our results show that similar genes or genes from the same families surprisingly are involved in biomineralization in both calcareous sponges and corals that produce different calcium carbonate polymorphs, calcite vs aragonite, respectively. This demonstrates that there are parallels between the formation of calcareous sponge spicules and coral skeleton formation at the proteome level. Furthermore, our results highlight how gene duplication and neofunctionalization of an original gene pool gave rise to dedicated biomineralization genes that could further differentiate to refine the biological control of the process. Single-cell expression data of calicoblastic cells in corals indicates that coral biomineralization may equally depend on small-scale spatial and temporal expression changes to fine-tune calcification, an aspect that requires further study and could help predict the responses of these reef-builders to global environmental changes.

# Materials and methods

## Key resources table

| Reagent type (species) or resource | Designation | Source or reference | Identifiers | Additional information |
| --- | --- | --- | --- | --- |
| Gene (*Sycon ciliatum*) | Calcarin 1–17 (Cal1–17) | This study | | De novo annotation of GenBank assembly GCA_964019385, available at https://zenodo.org/records/14755899, gene IDs provided in *Supplementary file 7* |

*Continued on next page*

*Continued*

| Reagent type (species) or resource | Designation | Source or reference | Identifiers | Additional information |
|---|---|---|---|---|
| Gene (*Sycon ciliatum*) | SciCarbonic anhydrase 1, Triactinin, Spiculin | *Voigt et al., 2014*; *Voigt et al., 2017* | | De novo annotation of GenBank assembly GCA_964019385, available at https://zenodo.org/records/14755899, gene IDs provided in *Supplementary file 7* |
| Biological sample (*Sycon ciliatum*) | DNA, RNA, tissue for *in situ* hybridization experiments | AWI Biologische Anstalt Helgoland, Germany | | Living specimens were shipped to Munich, Germany |
| Sequence-based reagent | PCR primers for generating probes for CISH | This study | PCR primers | Sequences of gene-specific primers for calcarin 1–8 are provided in *Appendix 1—table 4* |
| Sequence-based reagent | HCR-FISH probe sets | Molecular Instruments | | Probe sets consist of 20 pairs of probes per gene and were generated for Calcarin 1, Calcarin 2, Calcarin 3, Calcarin 7, Calcarin 8, Triactinin, Spiculin, SciCarbonic, andrase 1 by Molecular Instruments based on the de novo annotation of GenBank assembly GCA_964019385, available at https://zenodo.org/records/14755899, gene IDs provided in *Supplementary file 7* |
| Commercial assay or kit | RNA-Duet extraction kit | Zymo Research | Cat. # D7001 | Extraction of RNA |
| Commercial assay or kit | RNA 6000 Nano Kit | Agilent | Cat. # 5067-1511 | RNA extraction quality control |
| Commercial assay or kit | SENSE mRNA-Seq Library Prep Kit V2 | Lexogen | Cat. # 001.24 | Illumina library preparation |
| Commercial assay or kit | pCR4-TOPO cloning vector | Invitrogen | Cat. # K457502 | Used for probe generation in CISH |
| Commercial assay or kit | T3 polymerase | Promega | Cat. # P208C | Used for probe generation in CISH |
| Commercial assay or kit | T7 polymerase | Promega | Cat. # P207B | Used for probe generation in CISH |
| Commercial assay or kit | DIG RNA Labeling Mix | Roche | Cat. # 11277073910 | Used for generating DIG-labeled RNA probes |
| Commercial assay or kit | Fluorescein RNA Labeling Mix | Roche | Cat. # 11685619910 | Used for generating fluorescein-labeled RNA probes |
| Commercial assay or kit | NuPAGE 4–12% Bis-Tris Gel | Invitrogen | – | Preparation of proteins for mass spectrometry |
| Chemical compound, drug | NBT/BCIP Stock Solution | Roche | Cat. # 11681451001 | Substrate for CISH |
| Chemical compound, drug | FastRed Tablets | Roche | Cat. # 11496549001 | Substrate for CISH |
| Chemical compound, drug | EverBrite Hardset Mounting Medium | Biotum | Cat. # 23004 | Hardset antifade mounting medium with DAPI; used for mounting of tissue sections after HCR-FISH |
| Chemical compound, drug | Lysyl Endopeptidase (Lys-C), Mass Spectrometry Grade | FUJIFILM Wako Pure Chemical Corporation, USA | – | Used for in-gel digestion of proteins |
| Software, algorithm | Geneious | *Kearse et al., 2012* | RRID:SCR_010519 | Used for mapping trimmed reads to *Sycon* transcriptome |
| Software, algorithm | Salmon | *Patro et al., 2017* | RRID:SCR_017036, PMID:28263959 | Used for transcript quantification prior to DGE analysis |
| Software, algorithm | DESeq2 | *Love et al., 2014* | RRID:SCR_015687, DOI: 10.18 129/B9.bio c.DESeq2 | Version 1.42.1; used for analysis of differential gene expression between body parts and regeneration stages |
| Software, algorithm | WGCNA | *Langfelder and Horvath, 2008* | RRID:SCR_003302, PMID:19114008 | Version 1.72.5; WGCNA to identify gene modules associated with spicule formation |
| Software, algorithm | topGO | *Alexa and Rahnenfuhrer, 2023* | RRID:SCR_014798, DOI: 10.18129/B9.bioc.topGO | Version 2.54.0; GO-term enrichment analysis for genes overexpressed in osculum region and genes included in the 'midnightblue' module (WGCNA result) |
| Software, algorithm | REVIGO | *Supek et al., 2011* | RRID:SCR_005825, PMID:21789182 | Summarizing significantly enriched GO terms from GO analyses |
| Software, algorithm | TransPi | *Rivera-Vicéns et al., 2022* | PMID:35119207 | Nextflow-based pipeline for transcriptome assembly and annotation; used to reassemble raw reads and predict protein sequences for OrthoFinder analysis |
| Software, algorithm | BLASTp | *Camacho et al., 2009* | RRID:SCR_001010, PMID:20003500 | Used for homology search of galaxin-like proteins |
| Software, algorithm | OrthoFinder | *Emms and Kelly, 2019* | RRID:SCR_017118, PMID:31727128 | Version 2.5.5; orthogroup identification |
| Software, algorithm | MASCOT | Matrix Science Limited, UK, *Creasy et al., 1999* | RRID:SCR_014322, PMID:10612281 | Version 2.6.2; protein identification from LC-MS/MS spectra |
| Software, algorithm | Scaffold | Proteome Software Inc, Portland, USA | Version 5.01 | Available at : https://www.proteomesoftware.com/products/scaffold-5. Used for threshold filtering of identified proteins and visualization |
| Software, algorithm | Seurat | *Hao et al., 2024* | RRID:SCR_016341, DOI: 10.32614/CRAN.package.Seurat | Version 5.1.0 |
| Other | PepMap RSLC C18 | Thermo Scientific | | EASY-Spray column |
| Other | PepMap 100 C18 | Thermo Scientific | | Trap columns |

## Specimens, regeneration, and RNA-seq of body parts

Specimens of *S. ciliatum* were obtained from the AWI Biologische Anstalt Helgoland and sent alive to Munich. Here, specimens were transferred to glass Petri dishes and maintained in seawater for a few days with daily seawater changes. To verify the timing of the occurrence of spicules during the regeneration of *S. ciliatum*, we performed a regeneration experiment of the *S. ciliatum* as used for the regeneration RNA-seq dataset (*Soubigou et al., 2020*). We mechanically dissociated sponge cells through a 60 µm strainer, then incubated the cells in filtered seawater at 14°C in Petri dishes, changing the water every 1–2 days. The timing of whole-body regeneration stages and occurrence of spicules aligned with prior studies. After 30 days, the experiment ended, and we fixed some juvenile asconoid sponges for ISH.

For DGE analysis, we dissected three body parts from five living specimens (*Figure 1*): (1) The oscular region with mainly large diactine and tetractine spicules, (2) inner sponge wall, including the atrial skeleton (tetractines +diactines) and the proximal radial tube (triactines), (3) outer sponge wall, including the distal parts of the radial tube with triactines and tufts of curved diactines.

The body parts were instantly transferred into the lysis buffer of the RNA-Duet extraction kit (Zymo). We extracted RNA according to the manufacturer's instructions and confirmed its integrity with an Agilent Bioanalyzer using the RNA 6000 Nano Kit. We used the SENSE mRNA-Seq Library Prep Kit V2 for Illumina (Lexogen) to produce sequencing libraries. The 15 libraries were pooled and sequenced on a HiSeq Illumina Sequencer at the Gene Center of LMU. 100 bp paired raw reads were quality-checked with FastQC and trimmed to 92 bp to remove low-quality 3' bases.

## DGE analysis

Trimmed raw reads (*Appendix 1—table 1*) were mapped to a high-quality transcriptome of *S. ciliatum* (*Caglar et al., 2021*) using Geneious (*Kearse et al., 2012*). We omitted non-mapping reads to remove commensal sequences common in sponge tissues/samples. The trimmed and filtered reads were submitted to the European Nucleotide Archive (PRJEB78728). Raw reads from a whole-body regeneration experiment (*Soubigou et al., 2020*) were downloaded from the European Nucleotide Archive and processed identically (see *Appendix 1—table 2* for accession numbers). For mapping, we used the *S. ciliatum* transcriptome published in *Caglar et al., 2021*. This transcriptome showed better BUSCO (*Manni et al., 2021*) values than a transcriptome assembled using Trinity and the filtered reads from our experiments and was therefore preferred for mapping prior to DGE analysis with DESeq2 1.42.1 (*Love et al., 2014*). Gene and transcript counts for each filtered set were obtained with Salmon (*Patro et al., 2017*) and combined into count matrices for the body parts experiment and the regeneration data. For the body part dataset, we compared gene expression of the apical osculum region with increased spicule formation with gene expression of the more basally located regions of the sponge wall (inner and outer side). In the regeneration dataset, we identified the differentially expressed genes for two regeneration series between the initial spicule-free stages (days 1+2) and the subsequent stages that produce spicules (days 3–24). Differentially expressed genes ($|log2$-fold change$|\geq2$, padj$<0.01$) were identified, extracted from the reference transcriptome, and used in subsequent analysis.

## WGCNA

We combined both count datasets, filtered them to exclude low-count genes (genes with less than 10 samples with counts$>10$), and used them to construct a DESeq2 data set in R, followed by variance stabilizing transformation for normalization. We performed a WGCNA (1.72.5; *Langfelder and Horvath, 2012*; *Langfelder and Horvath, 2008*) to identify gene modules associated with spicule formation. For this purpose, a distinction was made between 'low spicule formation' for the body wall and the first stages of regeneration with no or only a few first diactines (days 0–4) compared to the other states (high spicule formation). The resulting modules were detected using the dynamic tree cut algorithm, and module eigengenes were calculated. A permutation test (n=1000) was conducted to assess the significance of modules regarding their association with conditions (low vs high spicule formation).

## GO-term enrichment analysis

Best hit proteins with e-values$\leq$e-05 were obtained by running BLASTx of *S. ciliatum* transcripts (*Caglar et al., 2021*) against the UniProt database. GO terms for these best hit proteins (e-values$\leq$e-05) were

retrieved from QuickGO (http://www.ebi.ac.uk/QuickGO/) using a custom Perl script. Transcript-associated GO terms were compiled by genes and used as input for GO-term enrichment analysis with topGO 2.54.0 (*Alexa and Rahnenfuhrer, 2023*) in R. This analysis was performed for genes overexpressed in the osculum region and genes included in the 'midnightblue' module identified by WGCNA. GO terms with fewer than 10 annotated genes were excluded. For each ontology (Molecular Function, Biological Process, and Cellular Component), gene lists were mapped to GO terms, and enrichment was tested using Fisher's exact test with the 'classic' algorithm. Significantly enriched GO terms ($p \leq 0.05$) were summarized using REVIGO (*Supek et al., 2011*) to provide representative terms. All scripts, input files, and parameters used in the analysis are provided in a GitHub repository.

## OrthoFinder and BLASTp

Raw RNA-seq data of several non-bilaterians was downloaded from SRA and reassembled using the TransPi pipeline (*Rivera-Vicéns et al., 2022*). The predicted proteins of these assemblies, of genomes and transcriptomes of calcareous sponges, sponges from other classes, octocorals, and stony corals were used as input to OrthoFinder to generate orthogroups (*Emms and Kelly, 2019*), focusing on species in which biomineralization genes are best studied: *A. millepora* (*Ramos-Silva et al., 2013*; *Reyes-Bermudez et al., 2009*; *Takeuchi et al., 2016*), *S. pistillata* (*Drake et al., 2013*; *Peled et al., 2020*; *Zoccola et al., 2015*), three octocoral species (*Conci et al., 2020*), the calcifying demosponge *Vaceletia* sp. (*Germer et al., 2015*), as well as some other high-quality sponge genomes (*Francis et al., 2023*; *Francis et al., 2017*; *Kenny et al., 2020*; *Shimizu et al., 2024*). Sequences of calcarins from Calcaronea were extracted from orthogroup sequences and aligned using MUSCLE (*Edgar, 2004*). We used BLASTp (*Camacho et al., 2009*) with *A. millepora* Galaxin (D9IQ16) with a maximum e-value of 1e-5 as the threshold to identify proteins similar to galaxins in the proteomes (*Supplementary file 6*).

## RNA ISH

We conducted RNA ISH to study the spatial and temporal expression of selected genes using standard chromogenic ISH and HCR-FISH. Fixation of specimens and chromogenic ISH followed published procedures (*Fortunato et al., 2012*). Further details about fixation, probe preparation, and HCR-FISH are provided in the Supplementary Information.

## Identification of spicules organic matrix proteins

Identification and analysis of organic matrix proteins from *S. ciliatum* spicules involved isolating 6 g of spicules with a solution of sodium hypochlorite, decalcifying them with acetic acid, and separating the acid-soluble and insoluble fractions, of which only the latter provided enough protein to be further analyzed. Proteomic analysis included gel electrophoresis, in-gel digestion, and LC-MS/MS. Proteins were identified using MASCOT (*Creasy et al., 1999*) with the predicted proteins from the *Sycon* transcriptome (PRJEB49276), and additional analyses were performed in Scaffold V5.01 (Proteome Software Inc, Portland, OR, USA). Detailed experimental procedures are available in Appendix 1.

## Genome analysis

We accessed the assembly of the *S. ciliatum* genome (GCA_964019385) of the Tree of Life Programme (https://www.sanger.ac.uk/programme/tree-of-life/, https://www.sanger.ac.uk/collaboration/aquatic-symbiosis-genomics-project/). Because we found the provided gene predictions were incomplete, we performed gene predictions using BRAKER3 installed on a public Galaxy server installation (https://usegalaxy.eu), using a HiSat2 (*Kim et al., 2019*) mapping of our filtered reads and the peptides from genomic scaffolds (*Fortunato et al., 2014*) as training data. For the stony corals, we obtained the annotated genomes from GenBank (*A. millepora* v2.1, GCF_013753865, *S. pistillata* v1.1, GCF_002571385). The genetic locations of biomineralization genes, additional carbonic anhydrases, and SLC4 transporters were identified using BLASTp. These coordinates were then extracted from the genome's GFF file.

## Structure predictions of calcarins

AlphaFold express (https://www.line-d.net/alphafold-express) was used to predict the structures of selected calcarins. AlphaFold predictions of *A. millepora* Galaxin (D9IQ16) and Galaxin 2 (B8UU51)

were downloaded from UniProt (*Bateman et al., 2021*) as pdb files. Model confidence for the structure predictions of the beta-hairpin structure was very high (pLDDT >90) in most cases, but it was generally low or lower for adjacent regions. We visualized predicted structures with the Mol* 3D viewer (*Sehnal et al., 2021*).

## Inspection of skeletal matrix protein expression in coral calicoblasts

To investigate fine-tuned changes in expression occurring in the coral *S. pistillata* calicoblasts, we examined publicly available single-cell sequencing datasets (*Levy et al., 2021*). Using the species' online cell atlas tool (https://sebe-lab.shinyapps.io/Stylophora_cell_atlas/), we visualized the expression of skeletal matrix proteins (*Peled et al., 2020*). This analysis revealed specific expression of 14 skeletal matrix proteins in calicoblast metacells of young coral polyps (*Appendix 2—figure 1*), whereas only very few adult coral calicoblast metacells expressed documented skeletal matrix proteins, indicating no or limited calcification activity (*Appendix 2—figure 6*). The raw UMI counts (Spis_polyp_sc_UMI_counts.RDS, Spis_adult_sc_UMI_counts.RDS) and the celltype assignments (Spis_polyp_cell_type_assignments.txt, Spis_coral_cell_type_assignments.txt) were downloaded from GitHub (https://github.com/sebepedroslab/Stylophora_single_cell_atlas; *Elek, 2025*). Single-cell RNA-seq data from cells expressing more than 100 genes were normalized and scaled using the LogNormalize and ScaleData functions of the Seurat R package version 5.1.0 (*Hao et al., 2024*), respectively, to adjust gene expression counts to 10,000 molecules per cell, and to center and scale each gene to a mean of zero and a variance of one. Normalized and scaled expression data of calicoblasts were visualized as heatmaps using pheatmap version 1.0.12.

## Acknowledgements

We gratefully acknowledge the financial support provided by the German Research Foundation (DFG, project VO 2238/1-1). We also thank the Darwin Tree of Life project (https://www.darwintreeoflife.org) and the Aquatic Symbiosis Genome Project (https://www.sanger.ac.uk/collaboration/aquatic-symbiosis-genomics-project/) for sequencing and providing the genomic resources of *S. ciliatum*. Some of the calculations were performed on the Galaxy server, which is partially funded by the Collaborative Research Centre 992 Medical Epigenetics (DFG grant SFB 992/1 2012) and the German Federal Ministry of Education and Research (BMBF grants 031 A538A/A538C RBC, 031L0101B/031L0101C de.NBI-epi, and 031L0106 de.STAIR [de.NBI]). We also thank Helmut Blum and Stefan Krebs from the Gene Center (Ludwig-Maximilians-Universität München) for performing the RNA-seq. Special thanks to Nora Dotzler and Lara Aust for their assistance with the HCR-FISH experiments and to Lara Aust for her work on the regeneration of *S. ciliatum*. Additionally, we appreciate the help of Bernhard Ruthensteiner (Zoologische Staatssammlung München) with the µCT scans. The English language was improved with the help of ChatGPT-4.0 (OpenAI; accessible at https://openai.com/blog/chatgpt).

## Additional information

### Funding

| Funder | Grant reference number | Author |
|---|---|---|
| Deutsche Forschungsgemeinschaft | 319411146 (VO 2238/1-1) | Oliver Voigt |

The funders had no role in study design, data collection and interpretation, or the decision to submit the work for publication.

### Author contributions

Oliver Voigt, Conceptualization, Data curation, Formal analysis, Supervision, Funding acquisition, Validation, Investigation, Visualization, Methodology, Writing – original draft, Project administration, Writing – review and editing, Obtained funding, designed the research approach, planned the

experimental work, performed in situ hybridization, analyzed RNA-seq data, submitted sequence data to ENA, prepared figures, wrote the initial draft, and contributed to manuscript editing and revision; Magdalena V Wilde, Data curation, Formal analysis, Investigation, Methodology, Data generation and analysis (proteome); Thomas Fröhlich, Resources, Formal analysis, Investigation, Writing – review and editing, Data generation and analysis (proteome); Benedetta Fradusco, Investigation, Data generation: RNA extraction, library preparation; Sergio Vargas, Investigation, Methodology, Support in data analysis, providing predicted proteins of transcriptomes; Gert Wörheide, Conceptualization, Resources, Writing – review and editing

### Author ORCIDs
Oliver Voigt https://orcid.org/0000-0001-8708-0872
Magdalena V Wilde http://orcid.org/0000-0002-1997-8768
Thomas Fröhlich https://orcid.org/0000-0002-4709-3211
Sergio Vargas https://orcid.org/0000-0001-8704-1339
Gert Wörheide https://orcid.org/0000-0002-6380-7421

### Ethics
No ethics approval is required for work on sponges.

Reviewer #1 (Public review): https://doi.org/10.7554/eLife.106239.3.sa1
Reviewer #2 (Public review): https://doi.org/10.7554/eLife.106239.3.sa2
Reviewer #3 (Public review): https://doi.org/10.7554/eLife.106239.3.sa3
Author response https://doi.org/10.7554/eLife.106239.3.sa4

---

## Additional files

### Supplementary files
MDAR checklist

Supplementary file 1. Biological process GO terms enriched in genes overexpressed in the oscular region of *S. ciliatum* (representative terms obtained from REVIGO).

Supplementary file 2. Proteins identified in the calcareous sponge spicule matrix. Proteins that are overexpressed in the oscular region (log2-fold change) are in bold, calcarins highlighted by yellow background.

Supplementary file 3. *Vaceletia* sp. skeletal proteins similar to *S. ciliatum* spicule matrix proteins.

Supplementary file 4. Genes overexpressed in the oscular region and included in meta module midnightblue with selected gene regulatory GO annotations.

Supplementary file 5. OrthoFinder results for orthogroups that include calcarins and other biomineralization genes.

Supplementary file 6. Galaxin BlastP hits in the proteomes used in OrthoFinder analysis and proteins and structures of PANTHER family 'PTHR34490'.

Supplementary file 7. Genomic position of biomineralization genes in *S. ciliatum*, *A. millepora*, and *S. pistillata*.

### Data availability
Sequencing data have been deposited in ENA under BioProject accession code PRJEB78728. Mass spectrometry proteomics data have been deposited to the ProteomeXchange Consortium via the PRIDE partner repository (*Perez-Riverol et al., 2019*) under accession code PXD060105. Scripts and input files for the RNA-Seq, GO-Term, and WGCNA analyses are available in a GitHub repository (https://github.com/PalMuc/CalcBiomin, copy archived at *Voigt, 2025*) and have been archived within a Zenodo repository https://doi.org/10.5281/zenodo.16786067. De novo gene predictions, detailed outputs from Alpha-Fold and the OrthoFinder-Analysis are provided along with a scaffold file for the proteins identified from the spicule matrix and can be accessed via Zenodo (https://doi.org/10.5281/zenodo.14755899). All data generated or analysed during this study are included in the manuscript and supporting files (Appendix 1 - table 1, Appendix 1 - table 2, Appendix 1 - table 3).

The following datasets were generated:

| Author(s) | Year | Dataset title | Dataset URL | Database and Identifier |
|---|---|---|---|---|
| Voigt et al. | 2025 | Calcareous sponge biomineralization | https://www.ebi.ac.uk/ena/browser/view/PRJEB78728 | European Nucleotide Archive, PRJEB78728 |
| Voigt O | 2025 | Supplementary data for "Genetic parallels in biomineralization of the calcareous sponge Sycon ciliatum and stony corals" | https://doi.org/10.5281/zenodo.14755899 | Zenodo, 10.5281/zenodo.14755899 |
| Fröhlich T | 2025 | Identification and analysis of organic matrix proteins from Sycon ciliatum spicules | https://proteomecentral.proteomexchange.org/cgi/GetDataset?ID=PXD060105 | ProteomeXchange, PXD060105 |
| Voigt O | 2025 | PalMuc/CalcBiomin: Updated README, minor bug fixes | https://doi.org/10.5281/zenodo.16786067 | Zenodo, 10.5281/zenodo.16786067 |

The following previously published dataset was used:

| Author(s) | Year | Dataset title | Dataset URL | Database and Identifier |
|---|---|---|---|---|
| Soubigou et al. | 2020 | Regeneration in sponge *Sycon ciliatum* partly mimics postlarval development | https://www.ebi.ac.uk/ena/browser/view/PRJNA628727 | European Nucleotide Archive, PRJNA628727 |

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

# Appendix 1

## Extended methods

### Chromogenic and HCR RNA ISH

We investigated the spatial and temporal expression of selected calcarins and previously characterized biomineralization genes through standard chromogenic RNA ISH and HCR-FISH. Living *S. ciliatum* were fixed using MOPS fixation buffer (100 mM MOPS, pH 7.5; 0.5 M sodium chloride; 2 mM MgSO₄; 4% paraformaldehyde, 0.05% glutaraldehyde) at 4°C overnight, followed by dehydration in 70% ethanol and storage at –20°C. Target genes were amplified from cDNA using gene-specific primers (*Appendix 1—table 4*) and cloned into pCR4-TOPO cloning vector (Invitrogen) with T3 or T7 initiation sequence adjacent to the insertion site. The plasmid was sequenced with vector-specific primers to determine the insert direction. From the plasmid, a template for in vitro transcription was generated by PCR with the gene-specific forward primer and a vector-specific primer (depending on the insertion direction). Antisense RNA probes for chromogenic ISH were generated by *in vitro* transcription using T3 or T7 RNA polymerase from plasmids or PCR products of the target genes. Probes were labeled with digoxigenin or fluorescein with a corresponding RNA labeling kit (Roche). Additional antisense RNA probes were available from a previous study (*Voigt et al., 2017*). The procedures for ISH followed published protocols (*Fortunato et al., 2012*; *Voigt et al., 2017*). We employed a two-probe detection system for double ISH, using NBT/BCIP (Roche) for the first probe and FastRed (Roche) for the second, and the documentation was performed using Leica microscopes (M165F, DMLB).

For HCR-FISH, gene-specific probe sets, each consisting of 20 pairs of gene-specific probes (*Appendix 1—table 5*) based on the target gene's transcript sequence, were designed and delivered by Molecular Instruments (https://www.molecularinstruments.com), which also delivered corresponding amplifiers, labeled with AF488, AF546, AF594, or AF647. HCR-FISH allowed us to visualize up to three target genes with different fluorochromes simultaneously. Tissue preparation for HCR-FISH mirrored the initial steps of ISH up to re-fixation and the subsequent washing steps, followed by whole-mount HCR-FISH as specified by the manufacturer. Post-HCR-FISH, tissues were left in 5x SSC overnight to dissolve remaining spicules that impede probe visualization due to their optical properties. We then mounted the tissue using EverBbrite Hardset (Biotum) for visualization or embedded it in Technovit (Kulzer) for sectioning. We used a LEICA Thunder Imager for fluorescent microscopy and photo documentation, using Z-stacking to extend the focal range. Background autofluorescence in thicker tissues was reduced using Leica's large volume computational image clearing implemented in Leica Application Suite (Leica). To improve accessibility for individuals with red/green color vision deficiency, the resulting HCR-FISH images of *Figure 3* were processed in ImageJ by splitting channels, applying cyan, magenta, or yellow lookup tables, converting each channel image to a RGB type image, and combining them using the Z-Project function with maximum intensity projection. A version with the original channel information preserved is provided in *Figure 3—figure supplement 3*.

### Identification of organic matrix proteins from spicules

*S. ciliatum* spicules were isolated by dissolving the surrounding tissue in a 5% sodium hypochlorite solution, followed by several rinses with Milli-Q water and air drying. Approximately 6 g of dried spicules were then decalcified overnight in 10% acetic acid at room temperature on an orbital shaker. This process separated the spicules into acid-soluble (ASM) and acid-insoluble matrices (AIM), then divided via centrifugation at 14,000 × *g* for 30 min. The ASM fraction was desalted using Amicon Ultrafiltration devices, and the AIM residues were thoroughly washed, and both fractions were subsequently stored at –20°C.

To prepare samples for LC-MS/MS, they were unthawed, briefly vortexed, and allowed to settle for 3–5 min 250 µl of the semitransparent phase was then transferred into Eppendorf tubes and centrifuged for 30 s at 5000 rpm and 4°C. The resulting pellet was washed twice with ultrapure mass-spectrometry grade water (MS-H₂O, Merck) and resuspended in 100 µl MS-H₂O. After sonication with a cup resonator (Bandelin), 2X Laemmli buffer was added, and the samples were incubated at 95°C for 5 min, followed by a second sonication. The prepared samples were vortexed and loaded onto a NuPAGE 4–12% Bis-Tris Gel (Invitrogen, USA). Electrophoresis was performed at 200 V until the gel pockets were empty, and the gel was subsequently stained overnight with Roti Blue staining solution (Roth, Germany). Protein-containing gel segments were excised, destained, and

washed with 50 mM NH₄HCO₃. The supernatant was discarded, and the gel pieces were reduced with 45 mM DTE for 30 min at 55°C, then alkylated with 100 mM iodoacetamide in the dark at room temperature. Following alkylation, the gel pieces were washed again, and a sequential in-gel digestion was executed using Lysyl Endopeptidase (Lys-C, 4 hr at 37°C, Mass Spectrometry Grade, FUJIFILM Wako Pure Chemical Corporation, USA) followed by an overnight digestion with 100 ng of sequencing grade modified trypsin (Promega, Germany) at 37°C. The resulting supernatants containing peptides were collected, and the peptides were extracted using 70% acetonitrile (ACN). These collected supernatants were pooled and then dried using a vacuum centrifuge (Bachofer, Germany).

LC-MS/MS analysis was performed using an Ultimate 3000 RSLC (Thermo Fisher Scientific) connected to a Q Exactive HF-X mass spectrometer (Thermo Fisher Scientific). Protein samples were loaded onto trap columns (PepMap 100 C18, 100 μm×2 cm, 5 μM particles, Thermo Scientific) using a flow rate of 5 μl/min (mobile phase: 0.1% formic acid and 1% acetonitrile in water). Liquid chromatography was performed with an EASY-Spray column (PepMap RSLC C18, 75 μm×50 cm, 2 μm particles, Thermo Scientific) and a 250 nl/min flow rate. Peptide samples were separated with a two-step gradient from 3% mobile phase B (0.1% formic acid in acetonitrile) to 25% mobile phase B in 30 min, followed by a ramp to 40% for 5 min (mobile phase A: 0.1% formic acid in water). The mass spectrometer was set to data-dependent acquisition mode, performing a maximum of 15 MS/MS spectra per survey scan.

MS/MS spectra were analyzed with MASCOT V2.6.2 (Matrix Science Limited, UK) (*Creasy et al., 1999*). The predicted proteins from the *S. ciliatum* transcriptome (PRJEB49276) served as a database to which we added a predicted protein sequence of Cal8 that we found as a second, untranslated protein encoded on transcript HBWS01158424. We further evaluated the results in Scaffold (v. 5.0.1, Proteome Software Inc, Portland, USA), setting the filters to a protein threshold of 1.0% FDR, the minimum number of peptides = 2, and a peptide threshold of 1.0% FDR. The mass spectrometry proteomics data have been deposited to the ProteomeXchange Consortium (http://proteomecentral.proteomexchange.org) via the PRIDE partner repository (*Perez-Riverol et al., 2019*) with the project accession PXD060105.

**Appendix 1—table 1.** Accession numbers or RNA-seq data generated for the body part dataset (BioProject PRJEB78728).

| Body part | Replicate specimen | Sample name | Accession |
|---|---|---|---|
| Osculum region | 1 | GW30948_OSC | ERR13472820 |
| Inner sponge wall | 1 | GW30948_IN | ERR13472821 |
| Outer sponge wall | 1 | GW30948_OUT | ERR13472822 |
| Osculum region | 3 | GW30951_OSC | ERR13472823 |
| Inner sponge wall | 3 | GW30951_IN | ERR13472824 |
| Outer sponge wall | 3 | GW30951_OUT | ERR13472825 |
| Osculum region | 4 | GW30956_OSC | ERR13472826 |
| Inner sponge wall | 4 | GW30956_IN | ERR13472827 |
| Outer sponge wall | 4 | GW30956_OUT | ERR13472828 |
| Osculum region | 2 | GW30957_OSC | ERR13472829 |
| Inner sponge wall | 2 | GW30957_IN | ERR13472830 |
| Outer sponge wall | 2 | GW30957_OUT | ERR13472831 |
| Osculum region | 5 | GW30959_OSC | ERR13472832 |
| Inner sponge wall | 5 | GW30959_IN | ERR13472833 |
| Outer sponge wall | 5 | GW30959_OUT | ERR13472834 |

**Appendix 1—table 2.** Accession for RNA-seq data from the regeneration experiment by *Soubigou et al., 2020*.

Set I and set II are two regeneration experiments followed for 24 days. Additional experiments of

the study were not used, because they did not include the first spicule-free stages.

| Experiment, time | Accession | Regeneration stage |
|---|---|---|
| Set I, day 1 | SRR11617503 | No spicules |
| Set II, day 1 | SRR11617504 | No spicules |
| Set I, day 2 | SRR11617505 | No spicules |
| Set II, day 2 | SRR11617506 | No spicules |
| Set I, days 3–4 | SRR11617507 | Primmorphs with spicules (diactines) |
| Set II, days 3–4 | SRR11617508 | Primmorphs with spicules (diactines) |
| Set I, days 6–8 | SRR11617513 | Ciliated chambers |
| Set II, days 6–8 | SRR11617514 | Ciliated chambers |
| Set I, days 10–14 | SRR11617519 | Choanoderm, expanding spongocoel, pinacoderm |
| Set II, days 10–14 | SRR11617520 | Choanoderm, expanding spongocoel, pinacoderm |
| Set I, days 16–18 | SRR11617523 | Osculum opens, porocytes form ostia |
| Set II, days 16–18 | SRR11617524 | Osculum opens, porocytes form ostia |
| Set I, days 21–24 | SRR11617526 | Juvenile |
| Set II, days 21–24 | SRR11617527 | Juvenile |
| Dissociated cells, day 0 | SRR11617528 | Adult |

**Appendix 1—table 3.** Source of the data used in the OrthoFinder analysis.

| Species | Accession (run, sample, study, genome) or other source | Protein predictions from |
|---|---|---|
| Porifera | | |
| Calcarea | | |
| Calacronea | | |
| *Grantia compressa* | SRR3417193 | transcriptome* |
| *Leuconia nivea* | SRR3417190 | transcriptome* |
| *Leucosolenia complicata* | http://compagen.unit.oist.jp/datasets.html | transcriptome* |
| *Sycon ciliatum* | GCA_964019385.1 | genome[†] |
| *Sycon ciliatum* (Bergen) | http://compagen.unit.oist.jp/datasets.html | genome |
| *Sycon ciliatum* (Helgoland) | ERP163002 (this study) | transcriptome* |
| Calcinea | | |
| *Clathrina coriacea* | SRR3417192 | transcriptome* |
| *Janusya* sp. | ERR5279461 | transcriptome* |
| *Leucetta chagosensis* | SRS8111786 | transcriptome* |
| Homocleromorpha | | |
| *Oscarella carmela* | http://compagen.unit.oist.jp/datasets.html | genome |
| Demospongiae | | |
| *Ephydatia muelleri* | https://bitbucket.org/EphydatiaGenome/ephydatiagenome/downloads/ | genome |
| *Tethya wilhelma* | https://bitbucket.org/molpalmuc/tethya_wilhelma-genome/ | genome |
| *Vaceletia* sp. | SRR4423080 | transcriptome* |

*Appendix 1—table 3 Continued on next page*

*Appendix 1—table 3 Continued*

| Species | Accession (run, sample, study, genome) or other source | Protein predictions from |
|---|---|---|
| Hexactinellida | | |
| *Acanthascus vastus* | **Francis, 2023**, genome version Avas v1.29, available at https://github.com/PalMuc/Aphrocallistes_vastus_genome/ | genome |
| *Vazella_pourtalesii* | https://doi.org/10.6084/m9.figshare.23799351 | |
| Cnidaria | | |
| Hexacorallia | | |
| *Acropora millepora* | GCF_013753865.1 | genome |
| *Acropora digitata* | GCF_000222465.1 | genome |
| *Stylophora pistillata* | GCF_002571385.2 | genome |
| Octocorallia | | |
| *Heliopora coerulea* | ERP120267 | transcriptome* |
| *Pinnigorgia flava* | ERP122203 | transcriptome* |
| *Tubipora musica* | ERR3026435 | transcriptome* |

*Re-assembled with TransPi.
†New gene prediction using BRAKER 3.

**Appendix 1—table 4.** Gene-specific primer sequences for generating *in situ* hybridization (ISH) probes.

| Gene | Primer name | Sequence (5'–3') |
|---|---|---|
| Calcarin 1 | SciCal1_fw | CACAACAATCCACGCAGCA |
| | SciCal1_rv | TCCACTGCAACAGCTCTCAG |
| Calcarin 2 | SciCal2_fw | GAACCATTCTGGGGAAAATGCC |
| | SciCal2_rv | TGGTTGGTATTGGCAGCTTCTC |
| Calcarin 3 | SciCal3_fw | AATACAACACGTCCAAACAGCG |
| | SciCal3_rv | CAAGACTTGCTTCTTTCCTGCC |
| Calcarin 4 | SciCal4_fw | GGAGAGTTCTTTTTCCCCGGAT |
| | SciCal4_rv | GCTTTGTTGTTGGTGAGACTCC |
| Calcarin 5 | SciCal5_fw | CTGCAACAACGAACCTATGCAA |
| | SciCal5_rv | CATCTGCATACCAGGCATCATG |
| Calcarin 6 | SciCal6_fw | CGTGGGGAGAATACTTCACCAA |
| | SciCal6_rv | GACGCGACATTGTTCAATCCAA |
| Calcarin 7 | SciCal7_fw | GCGAGAAGGCTAGCTATCATGT |
| | SciCal7_rv | CTCTTGGAAAGCGCATACATGG |
| Calcarin 8 | SciCal8_fw | AGAAGGAGACGCTAGTACTGGT |
| | SciCal8_rv | TACGGATTGTAGATGTCGGCAC |

**Appendix 1—table 5.** Hairpin chain reaction fluorescence *in situ* hybridization (HCR-FISH) probe sets, each consisting of 20 pairs of gene-specific probes with specific split HCR initiators. Visualization of co-expressed genes requires each target probe set to have a different HCR initiator.

| Target gene | HCR initiator |
|---|---|
| Triactinin | B1 |
| Spiculin | B3 |

*Appendix 1—table 5 Continued on next page*

*Appendix 1—table 5 Continued*

| Target gene | HCR initiator |
|---|---|
| Calcarin 1 | B2 |
| Calcarin 2 | B1 |
| Calcarin 3 | B1 |
| Calcarin 7 | B1 |
| Calcarin 8 | B1 |
| *S. ciliatum* carbonic anhydrase 1 | B1 |

# Appendix 2—Supplementary figures

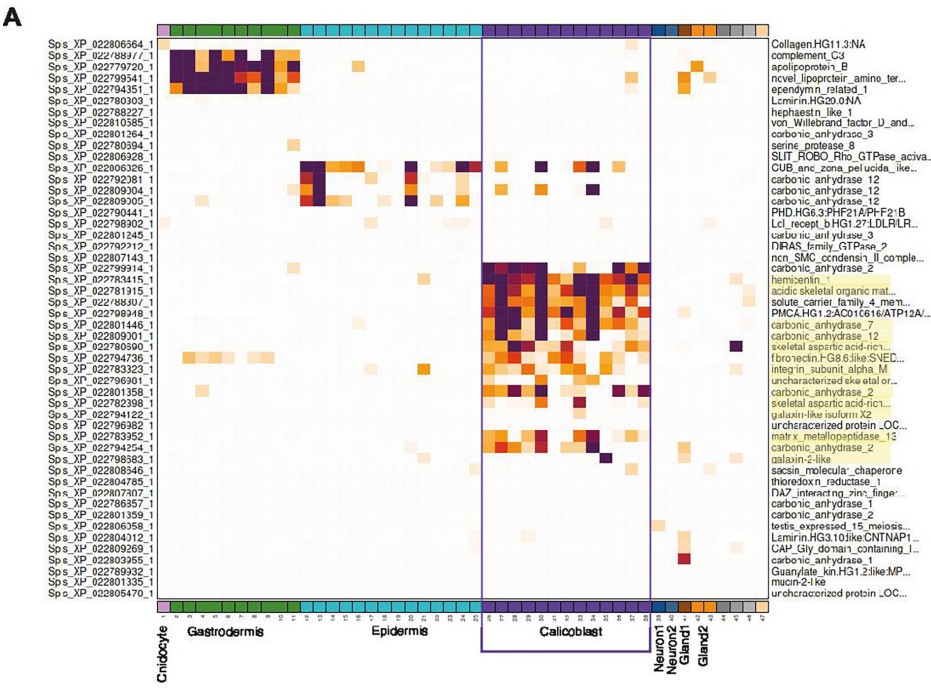

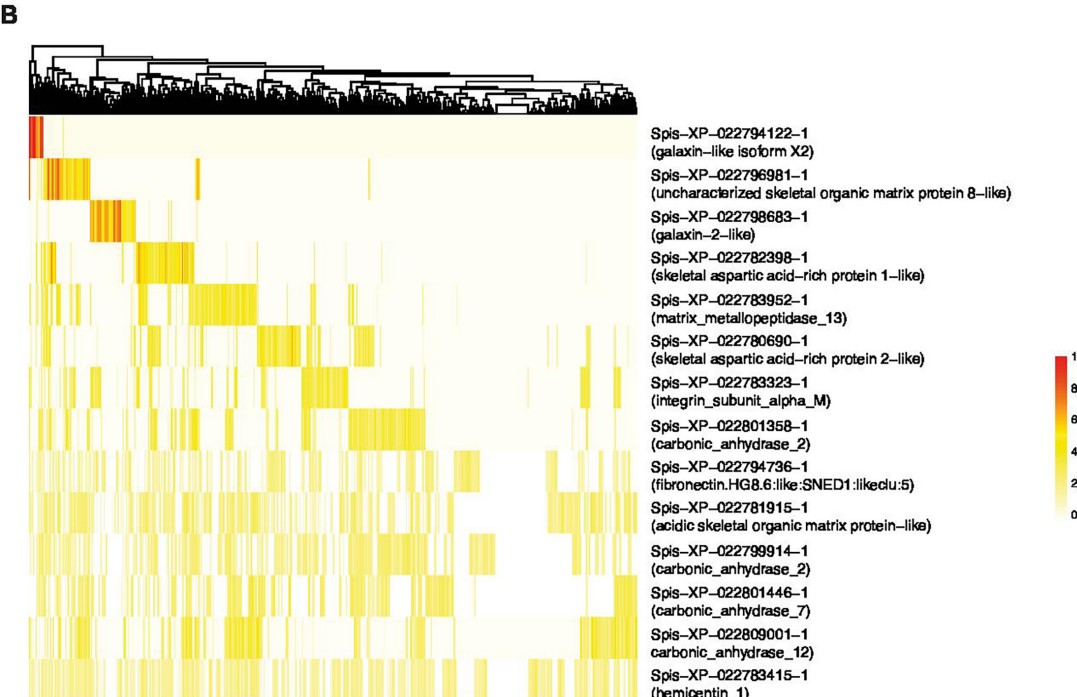

**Appendix 2—figure 1.** Expression of skeletal organic matrix (SOM) proteins in cells of young *S. pistillata* polyps. (**A**) Fourteen of the known SOM proteins (**Peled et al., 2020**) are specifically overexpressed in calicoblast metacells. Graph obtained from https://sebe-lab.shinyapps.io/Stylophora_cell_atlas/. (**B**) Normalized and scaled expression of 980 calicoblast cells show that several secreted SOM proteins are exclusively expressed by different calicoblast cells, suggesting a spatiotemporal expression regulation as observed in calcareous sponges.

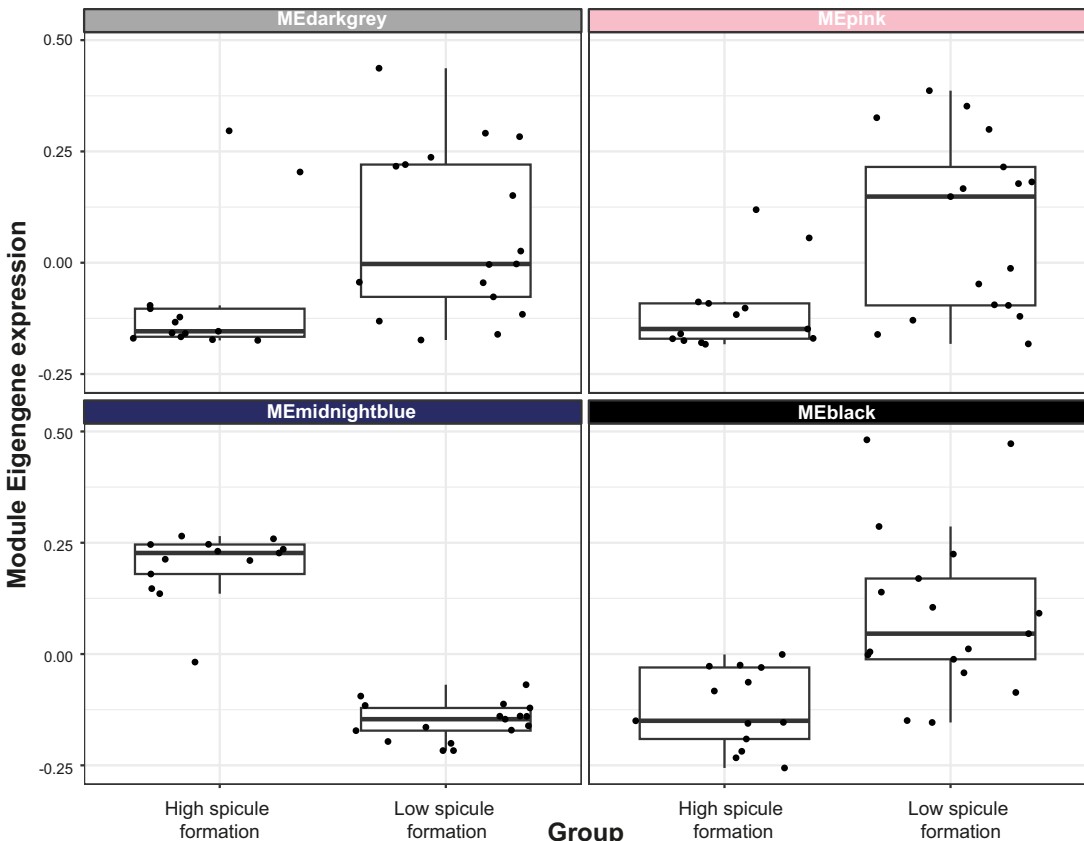

**Appendix 2—figure 2.** Changes in module eigengene (ME) expression between low spicule formation and high spicule formation transcriptomes. Each dot represents one RNA-seq library (low spicule formation: n = 17; high spicule formation: n = 13). Boxes indicate the interquartile range (IQR), whiskers extend to 1.5 × IQR, and the horizontal line marks the median. Most known biomineralization effector genes occur in MEmidnightblue.

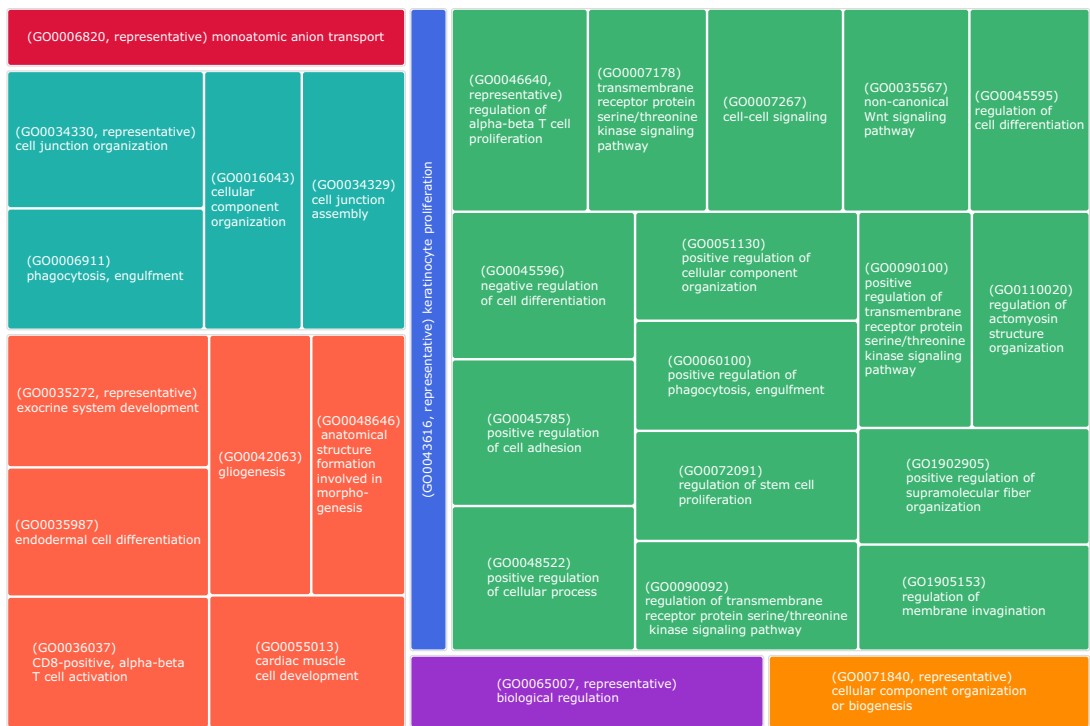

**Appendix 2—figure 3.** Enriched biological process GO terms (Treemap from REVIGO) of genes in meta module MEmidnightblue.

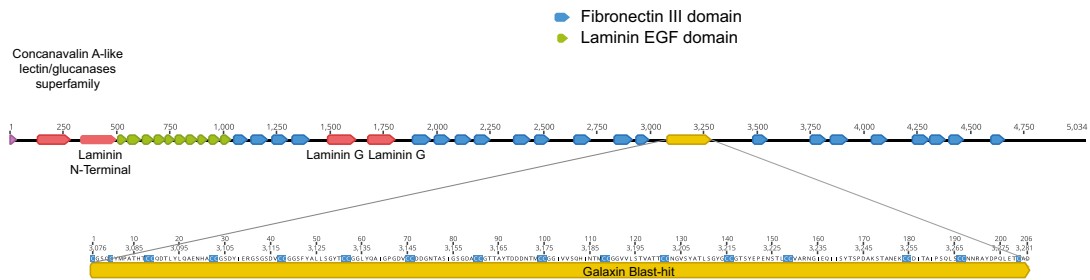

**Appendix 2—figure 4.** Domain structure of the only glass sponge (*Vazella pourtalesii*) protein with a blast hit for a Galaxin query (*Supplementary file 6*).

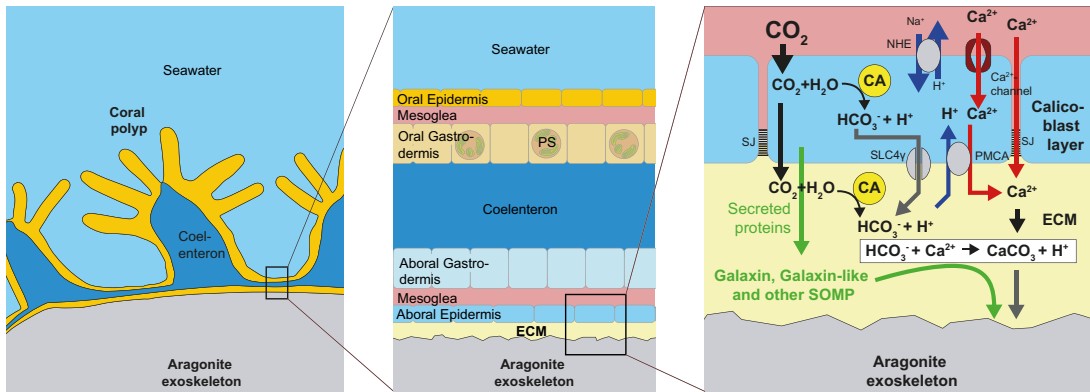

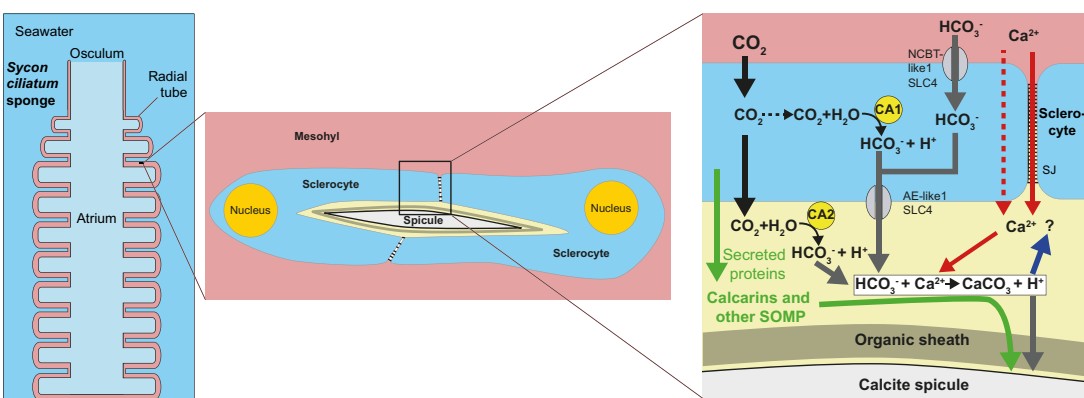

**Appendix 2—figure 5.** Schematic overview of skeleton formation in stony corals (top) and calcareous sponges (bottom). In stony corals, the skeleton is an extracellular exoskeleton deposited beneath the calicoblastic cell layer (the aboral epidermis), within a semi-isolated compartment known as the extracellular calcifying medium (ECM). Carbonic anhydrases in the cytosol and ECM catalyze the conversion of $CO_2$ to $HCO_3^-$, supplying inorganic carbon for calcification. The AE-like SLC4$\gamma$ transporter, localized to the apical membrane of calicoblastic cells, exports $HCO_3^-$ into the ECM. $Ca^{2+}$ reaches the ECM via both paracellular and transcellular pathways. In the paracellular route, $Ca^{2+}$ diffuses through septate junctions between calicoblastic cells. The transcellular route involves $Ca^{2+}$ influx channels on the basolateral membrane and plasma membrane $Ca^{2+}$-ATPases (PMCAs) on the apical membrane, which export $Ca^{2+}$ into the ECM while simultaneously importing protons ($H^+$). To maintain intracellular pH, $H^+$ is extruded from calicoblastic cells via $Na^+/H^+$ exchangers (NHEs). In the ECM, $Ca^{2+}$ and $HCO_3^-$ react to form calcium carbonate ($CaCO_3$), the mineral phase of the skeleton. Calicoblastic cells also secrete skeletal organic matrix proteins (SOMPs), such as galaxin and galaxin-like proteins, into the ECM, where they likely modulate crystal nucleation and growth. In calcareous sponges, the skeleton consists of calcite spicules formed by sclerocytes located in the mesohyl. Each spicule develops within an extracellular calcifying space enclosed by at least two sclerocytes (e.g. in diactine formation), which are connected by septate junctions (SJ) that seal the compartment. Inside this space, the growing spicule is surrounded by an organic sheath. Carbonic anhydrases, including mitochondrial (e.g. *S. ciliatum* CA1) or cytosolic forms, catalyze the conversion of $CO_2$ to $HCO_3^-$, providing inorganic carbon for calcification. Two sclerocyte-specific SLC4 family $HCO_3^-$ transporters, AE-like1 and NCBT-like1, mediate $HCO_3^-$ export into and import from the calcifying space, respectively (note that their apical and basolateral localization as depicted here is speculative). $Ca^{2+}$ is thought to enter the calcifying space via the paracellular route through junctional spaces between sclerocytes. Components of a transcellular $Ca^{2+}$ pathway have not yet been characterized. SOMPs, such as calcarins, are secreted into the calcifying space, where they likely influence the biomineralization and get incorporated into the calcite spicule.

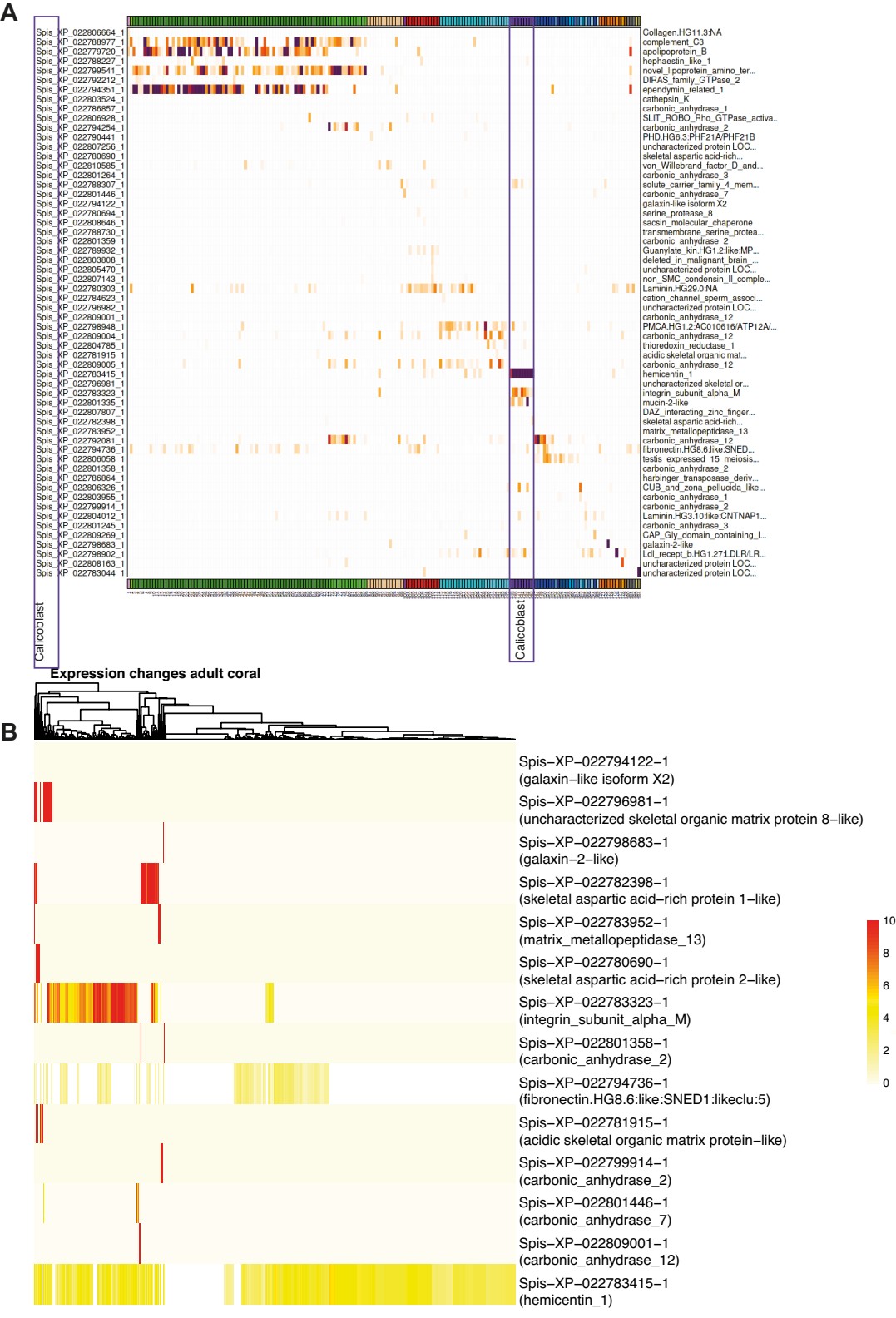

**Appendix 2—figure 6.** Expression of skeletal organic matrix (SOM) proteins in adult *S. pistillata* corals. (**A**) Most calicoblast metacells did not express known SOM proteins (*Peled et al., 2020*). Graph obtained from https://sebe-lab.shinyapps.io/Stylophora_cell_atlas/. (**B**) Normalized and scaled expression of the 14 SOM proteins specific to polyp calicoblasts (*Appendix 2—figure 1*) in 896 calicoblast cells of adult corals.

