## [Editor Report · eLife Assessment]

This **important** paper reports the discovery of calcarins, a protein family that seems to be involved in calcification in the calcareous sponge *Sycon ciliatum*, significantly enhancing our understanding of the molecular and cellular mechanisms underlying spicule formation in sponges and the evolution of carbonate biomineralization. The conclusions are supported by **compelling** evidence based on an integrated analysis that combines transcriptomics, genomics, proteomics, and precise *in situ* hybridization. These findings will be of broad interest to cell biologists, biochemists, and evolutionary biologists.

---

## [Referee Report · Reviewer #1 (Public review)]

To elucidate the mechanisms and evolution of animal biomineralization, Voigt et al. focused on the sponge phylum-the earliest branching extant metazoan lineages exhibiting biomineralized structures-with a particular emphasis on deciphering the molecular underpinnings of spicule formation. This study centered on calcareous sponges, specifically Sycon ciliatum, as characterized in previous work by Voigt et al. In S. ciliatum, two morphologically distinct spicule types are produced by set of two different types of cells that secrete extracellular matrix proteins, onto which calcium carbonate is subsequently deposited. Comparative transcriptomic analysis between a region with active spicule formation and other body regions identified 829 candidate genes involved in this process. Among these, the authors focused on the calcarine gene family, which is analogous to the Galaxins, the matrix proteins known to participate in coral calcification. The authors performed three-dimensional structure prediction using AlphaFold, examined mRNA expression of Calcarin genes in spicule-forming cell types via in situ hybridization, conducted proteomic analysis of matrix proteins isolated from purified spicules, and carried out chromosome arrangement analysis of the Calcarin genes. Based on these analyses, it was revealed that the combination of Calcarin genes expressed during spicule formation differs between the founder cells-responsible for producing diactines and triactines-and the thickener cells that differentiate from them, underscoring the necessity for precise regulation of Calcarin gene expression in proper biomineralization. Furthermore, the observation that 4 Calcarin genes are arranged in tandem arrays on the chromosome suggests that two rounds of gene duplication followed by neofunctionalization have contributed to the intricate formation of S. ciliatum spicules. Additionally, similar subtle spatiotemporal expression patterns and tandem chromosomal arrangements of Galaxins during coral calcification indicate parallel evolution of biomineralization genes between S. ciliatum and aragonitic corals.

Strength:

The study presents detailed and convincing insights that point to parallel evolution of biomineralization in calcitic sponges and corals. This is supported by a comprehensive analysis employing a wide range of experimental approaches including protein tertiary structure predictions, gene expression profiling during calcification (RNA seq and Whole-mount in situ hybridization), and chromosomal sequence analysis.

An integrative research approach, encompassing transcriptomic, genomic, and proteomic analyses as well as detailed FISH.

High-quality FISH images of Calcarin genes, along with a concise summary clearly illustrating their expression patterns, is appreciated.

It was suggested that thickener cells originate from founder cells. To the best of my knowledge, this is the first study to demonstrate trans-differentiation of sponge cells based on the cell-type specific gene expression, as determined by in situ hybridization.

Overall, this is a high-quality piece of work that proposes a compelling scenario for biomineralization.

Weaknesses:

I found no significant weakness in this manuscript.

Comments on revisions:

The authors have addressed all of the questions and recommendations from the prior review.

---

## [Referee Report · Reviewer #2 (Public review)]

Summary:

This paper reports on the discovery of calcarins, a protein family that seems involved in calcification in the sponge Sycon ciliatum, based on specific expression in sclerocytes and detection by mass spectrometry within spicules. Two aspects stand out: (1) the unexpected similarity between Sycon calcarins and the galaxins of stony corals, which are also involved in mineralization, suggesting a surprising, parallel co-option of similar genes for mineralization in these two groups; (2) the impressively cell-type-specific expression of specific calcarins, many of which are restricted to either founder or thickener cells, and to either diactines, triactines, or tetractines. The finding that calcarins likely diversified at least partly by tandem duplications (giving rise to gene clusters) is a nice bonus.

Strengths:

I enjoyed the thoroughness of the paper, with multiple lines of evidence supporting the hypothesized role of calcarins: spatially and temporally resolved RNAseq, mass spectrometry, and whole-mount in situ hybridization using CISH and HCR-FISH (the images are really beautiful and very convincing). The structural predictions and the similarity to galaxins are very surprising and extremely interesting, as they suggest parallel evolution of biomineralization in sponges and cnidarians during the Cambrian explosion by co-option of the same "molecular bricks".

Weaknesses:

I did not detect any major weakness, beyond those inherent to working with sponges (lack of direct functional inhibition of these genes) or with fast-evolving gene families with complex evolutionary histories (lack of a phylogenetic tree that would clarify the history of galaxins/calcarins and related proteins).

Comments on revisions:

I am fully satisfied with the revision, and notably with the new Figure 3 which is now extremely informative and readable. Congratulations on a job well done.

---

## [Referee Report · Reviewer #3 (Public review)]

Summary:

Voigt et al. present a comprehensive study exploring the molecular mechanisms and evolution of biomineralization in the calcareous sponge Sycon ciliatum. Using a multi-omics approach, including comparative transcriptomics, proteomics, genomic analyses, and high-resolution in situ hybridization, the authors identify 829 candidate biomineralization genes, with a special focus on the calcarin gene family. These calarains, structurally analogous to galaxin in stony corals, show cell-type- and spicule-type-specific expression patterns, revealed through meticulous FISH imaging. Chromosomal analysis further uncovers that several calcarin genes are arranged in tandem arrays, suggesting diversification via gene duplication and neofunctionalization. Notably, the study finds striking parallels between the calcarins of S. ciliatum and galaxins of aragonitic corals in terms of gene arrangement, tertiary structure predictions, and expression dynamics, pointing to a remarkable case of parallel evolution during the emergence of biomineralized skeletons in early metazoans.

Strengths:

The study is methodologically robust, integrating transcriptomic, proteomic, and genomic data with detailed cell biological analysis.

High-quality, carefully annotated FISH images convincingly demonstrate the spatial expression patterns of calcarins.

Novel evidence of sponge cell trans-differentiation is presented through cell-type-specific gene expression.

The comparative perspective with coral galaxins is well-executed and biologically insightful, supported by structural predictions and chromosomal data.

Figures and supplementary materials are thoughtfully revised for clarity and accessibility, addressing reviewer feedback.

Weaknesses:

Direct functional validation of calcarin roles in biomineralization is lacking, a limitation acknowledged by the authors and inherent to sponge models.

The evolutionary history of calcarins and galaxins remains only partially resolved due to challenges in reconstructing phylogenies of fast-evolving gene families.

Some initial figure annotations and definitions (e.g., "radial tube") required clarification, although these were addressed in revision.

Overall, the work significantly advances our understanding of biomineralization´s molecular basis and its parallel evolution in early diverging metazoans.

Comments on revisions:

I would like to thank the authors for addressing all my comments/suggestions. I am OK with the revised version of the manuscript

---

## [Author Response]

The following is the authors’ response to the original reviews.

**Reviewer #1 (Public review):**
To elucidate the mechanisms and evolution of animal biomineralization, Voigt et al. focused on the sponge phylum - the earliest branching extant metazoan lineages exhibiting biomineralized structures - with a particular emphasis on deciphering the molecular underpinnings of spicule formation. This study centered on calcareous sponges, specifically Sycon ciliatum, as characterized in previous work by Voigt et al. In S. ciliatum, two morphologically distinct spicule types are produced by a set of two different types of cells that secrete extracellular matrix proteins, onto which calcium carbonate is subsequently deposited. Comparative transcriptomic analysis between a region with active spicule formation and other body regions identified 829 candidate genes involved in this process. Among these, the authors focused on the calcarine gene family, which is analogous to the Galaxins, the matrix proteins known to participate in coral calcification. The authors performed three-dimensional structure prediction using AlphaFold, examined mRNA expression of Calcarin genes in spiculeforming cell types via in situ hybridization, conducted proteomic analysis of matrix proteins isolated from purified spicules, and carried out chromosome arrangement analysis of the Calcarin genes.Based on these analyses, it was revealed that the combination of Calcarin genes expressed during spicule formation differs between the founder cells-responsible for producing diactines and triactinesand the thickener cells that differentiate from them, underscoring the necessity for precise regulation of Calcarin gene expression in proper biomineralization. Furthermore, the observation that 4 Calcarin genes are arranged in tandem arrays on the chromosome suggests that two rounds of gene duplication followed by neofunctionalization have contributed to the intricate formation of S. ciliatum spicules. Additionally, similar subtle spatiotemporal expression patterns and tandem chromosomal arrangements of Galaxins during coral calcification indicate parallel evolution of biomineralization genes between S. ciliatum and aragonitic corals.Strengths:(1) An integrative research approach, encompassing transcriptomic, genomic, and proteomic analyses as well as detailed FISH.(2) High-quality FISH images of Calcarin genes, along with a concise summary clearly illustrating their expression patterns, is appreciated.(3) It was suggested that thickener cells originate from founder cells. To the best of my knowledge, this is the first study to demonstrate trans-differentiation of sponge cells based on the cell-typespecific gene expression, as determined by in situ hybridization.(4) The comparison between Calcarins of Calcite sponge and Galaxins of aragonitic corals from various perspective-including protein tertiary structure predictions, gene expression profiling during calcification, and chromosomal sequence analysis to reveal significant similarities between them.

We thank the reviewer for this assessment.

(1) The conclusions of this paper are generally well supported by the data; however, some FISH images require clearer indication or explanation.

We have modified Fig. 3 by including some insets indicating the depicted part of the sponge body and to change the color-scheme as suggested by reviewer3 for the FISH images. In accordance to the following comment, we decided to remove single-channel views in Fig. 3 A.

(2) Figure S2 (B, C, D): The fluorescent signals in these images are difficult to discern. If the authors choose to present signals at such low magnification, enhancing the fluorescence signals would improve clarity. Additionally, incorporating Figure S2A as an inset within Figure S2E may be sufficient to convey the necessary information about signal localization.

We changed the figure according to the suggestions.

(3) Figure S3A: The claim that Cal2-expressing spherical cells are closely associated with the choanoderm at the distal end of the radial tube is difficult to follow. Are these Cal2-expressing spherical cells interspersed among choanoderm cells, or are they positioned along the basal surface of the choanoderm? Clarifying their precise localization and indicating it in the image would strengthen the interpretation.

In the figure, the view is on the choanoderm that lines the inner surface of the radial tube. Our interpretation is that the spherical cells are positioned at the basal surface of the choanoderm. We updated Fig. S3, which now includes another view to support our interpretation and also indicate some choanocytes.

(4) To further highlight the similarities between S.ciliatum and aragonitic corals in the molecular mechanisms of calcification, consider including a supplementary figure providing a concise depiction of the coral calcification process. This would offer valuable context for readers.

We considered this suggestion, and have included such a supplementary figure (Fig. S9).

**Reviewer #2 (Public review):**
Summary:This paper reports on the discovery of calcarins, a protein family that seems involved in calcification in the sponge Sycon ciliatum, based on specific expression in sclerocytes and detection by mass spectrometry within spicules. Two aspects stand out: (1) the unexpected similarity between Sycon calcarins and the galaxins of stony corals, which are also involved in mineralization, suggesting a surprising, parallel co-option of similar genes for mineralization in these two groups; (2) the impressively cell-type-specific expression of specific calcarins, many of which are restricted to either founder or thickener cells, and to either diactines, triactines, or tetractines. The finding that calcarins likely diversified at least partly by tandem duplications (giving rise to gene clusters) is a nice bonus.Strengths:I enjoyed the thoroughness of the paper, with multiple lines of evidence supporting the hypothesized role of calcarins: spatially and temporally resolved RNAseq, mass spectrometry, and whole-mount in situ hybridization using CISH and HCR-FISH (the images are really beautiful and very convincing). The structural predictions and the similarity to galaxins are very surprising and extremely interesting, as they suggest parallel evolution of biomineralization in sponges and cnidarians during the Cambrian explosion by co-option of the same "molecular bricks".Weaknesses:I did not detect any major weakness, beyond those inherent to working with sponges (lack of direct functional inhibition of these genes) or with fast-evolving gene families with complex evolutionary histories (lack of a phylogenetic tree that would clarify the history of galaxins/calcarins and related proteins).

We thank the reviewer for this assessment and the detailed comments be addressed below.

**Reviewer #3 (Public review):**
Summary:The study explores the extent to which the biomineralization process in the calcitic sponge Sycon ciliatum resembles aragonitic skeleton formation in stony corals. To investigate this, the authors performed transcriptomic, genomic, and proteomic analyses on S. ciliatum and examined the expression patterns of biomineralization-related genes using in situ hybridization. Among the 829 differentially expressed genes identified in sponge regions associated with spicule formation, the authors focused on calcarin genes, which encode matrix proteins analogous to coral galaxins. The expression patterns of calcarins were found to be diverse but specific to particular spicule types. Notably, these patterns resemble those of galaxins in stony corals. Moreover, the genomic organization of calcarine genes in S. ciliatum closely mirrors that of galaxin genes in corals, suggesting a case of parallel evolution in carbonate biomineralization between calcitic sponges and aragonitic corals.Strengths:The manuscript is well written, and the figures are of high quality. The study design and methodologies are clearly described and well-suited to addressing the central research question. Particularly noteworthy is the authors´ integration of various omics approaches with molecular and cell biology techniques. Their results support the intriguing conclusion that there is a case of parallel evolution in skeleton-building gene sets between calcitic sponges and aragonitic corals. The conclusions are well supported by the data and analyses presented.Weaknesses:The manuscript is strong, and I have not identified any significant weaknesses in its current form.

We thank the reviewer for the insight and addressed the detailed comments below.

**Reviewer #1 (Recommendations for the authors):**
The description of the region "radial tube" is unclear. Please define and explain it at its first mention in the manuscript, and, if possible, refer to the appropriate figure(s) (e.g., Figure 1A).

We now explain radial tubes at the beginning of the results and added a label in figure 1A. “Sycon ciliatum is a tube-shaped sponge with a single apical osculum and a sponge wall of radial tubes around the central atrium (Fig. 1A). The radial tubes are internally lined with choanoderm, which forms elongated chambers in an angle of approximately 90° to the tube axis”.

**Reviewer #2 (Recommendations for the authors):**
Scientific suggestions:(1) Page 13: "Despite their presence in the same orthogroups, the octocoral and stony coral proteins were only distantly related to the calcareous sponge calcarins (e.g., 12-24% identity between octocoral and calcareous sequences in orthogroup Cal 2-4-6), resulting in poor alignment. Their homology to calcarins, therefore, remains to be determined." Could 3D structures of these coral proteins be predicted with AlphaFold to substantiate (or nuance) the comparison with calcarins?

We run additional alphafold predictions for two octocoral and two scleractinian galaxins. A galaxin-like sequence from Pinnigorgia flava was only a short fragment and therefore we did not attempt any structure predictions. The result shows that the octocoral galaxin-like proteins show some structural similarity (12 beta-harpins), while the scleractinian galaxin-like proteins differ from the sponge counterparts of the same orthogroup. We added this information to the results and in the new Fig. S7.

Minor improvements to the text:(1) Page 7 : "The expression of Cal1 to Cal8 was investigated using chromogenic in situ hybridization (CISH) and hairpin-chain reaction fluorescence in situ hybridization (HCR-FISH), confirming their presence in sclerocytes." - Figure 3 should be cited here.

We refer to the figure now.

(2) Page 8-9: "Cal6 expression mirrors that of Cal2, occurring in rounded cells at the distal tip of radial tubes and in a ring of cells around the oscular ring." - Please cite a figure here.

We refer now to Fig. 3K

(3) Page 11-12: Please define eigengene, this term is not necessarily common knowledge.

We provide now a short definition in this sentence: “ The analysis provided eight meta-modules, of which four showed significant changes in expression module eigengenes —summary profiles that capture the overall expression pattern of each module— between samples with high spicule formation context (osculum region and regeneration stages older than four days) and samples with low spicule formation (sponge-wall and early regeneration stages until day 3-4) (Fig. S5).”

(4) Page 13: "Species without skeletons, such as the cnidarians Hydra, Actinia, Exaiptasia, and Nematostella, also possess galaxin-like proteins." This is too concise - can you explain what evidence was used? PANTHER, AlphaFold, OrthoFinder, Blastp...?

The evidence used is from PANTHER, and we enhanced clarification of this by modifying the last sentence of the section.

(5) Page 20: "We have identified calcarins, galaxin-like proteins, as crucial components of the biomineralization toolkit in calcareous sponges." I'm not sure you showed they are crucial (this would require functional evidence). Perhaps "novel" components or some other adjective would fit better.

We changed the adjective to “novel”.

Suggestions for the figures:(1) Figure 1A: radial tubes should be labelled.

A label was added.

(2) Figure 3 is beautiful but hard to parse. The name of all markers should be written on each panel (notably B, C, and D) and ideally placed in a consistent position (top right corner?) so that the reader's eye doesn't have to look for them anew in each panel. Consider depicting the same gene with the same color in all panels if possible (confocal imaging gives virtual colors anyway, there's no reason to be bound to the real-life color of the fluorophores used - if that was the original intent). Finally, the red/green color scheme is not colorblind-readable, so please consider switching to another scheme (white/cyan/magenta, for example).

We have updated the figure according to the suggestions. The names of all markers are now included on each panel. Placing them in the upper right corner was not feasible for all panels, so we adjusted their placement as needed. Reoccurring genes are shown in the same color where possible. To improve accessibility for individuals with red/green color vision deficiency, we adopted a cyan/magenta/yellow color scheme. Each HCR-FISH image was processed in ImageJ by splitting the image into channels, applying cyan, magenta, or yellow lookup tables, converting each channel to RGB, and then stacking and blending them using the Z-Project function with maximum intensity projection. Since the original channel information is not preserved after this processing, we provide the original red/green/blue version of the figure in the supplementary material in Fig S11. Additionally, we added small sketches of Figure 1A to indicate the sponge body regions depicted, where relevant.

(3) Figure S3: the blue staining is not explained. It is also unclear where choanocytes are - could individual choanocytes be indicated with arrows or lines?

We added the information to the figure legend. The blue channel shows “Autofluorescence detected with the Leica TXR filter (approx. 590–650 nm), included to help distinguish true signal from background autofluorescence observed in the FITC channel (used for Spiculin detection).”

**Reviewer #3 (Recommendations for the authors):**
I have no major concerns about the manuscript - only minor edits and comments, which are listed below:(1) On page 13, the authors refer to Figure S8; however, I believe this should be Figure S7.

We now refer to the correct Figure. Because of introducing a new Fig. S7, now the correct reference is Fig. S8.

(2) On page 16, please correct "Spciulin" to "Spiculin".

Now corrected.

(3) On page 17, there are two commas following "(Sycon)"; please remove one.

Corrected.

(4) In the Data Accessibility section, none of the provided links appear to work. Please ensure all links are functional.

We apologize for this oversight and now provide working links.

(5) In Figure 3, the description of panel L is missing from the figure legend.

We added the description of this panel.

(6) On page 39, change "Fig. 4" to "Figure 4" to maintain consistency throughout the manuscript.

Changed.

(7) Figure S7 is not cited in the main text. Please, address this.

Corrected (see above at point 1)

(8) In the legend for Table S2, the reference to Soubigou et al. (3) is incorrect, as it is not listed in the SI reference section. Please correct this.

Soubigou et al. (2020) is now included in the SI reference list.